# Spatially integrative metrics reveal hidden vulnerability of microtidal salt marshes

Neil K. Ganju[1], Zafer Defne[1], Matthew L. Kirwan[2], Sergio Fagherazzi[3], Andrea D'Alpaos[4] & Luca Carniello[4]

Salt marshes are valued for their ecosystem services, and their vulnerability is typically assessed through biotic and abiotic measurements at individual points on the landscape. However, lateral erosion can lead to rapid marsh loss as marshes build vertically. Marsh sediment budgets represent a spatially integrated measure of competing constructive and destructive forces: a sediment surplus may result in vertical growth and/or lateral expansion, while a sediment deficit may result in drowning and/or lateral contraction. Here we show that sediment budgets of eight microtidal marsh complexes consistently scale with areal unvegetated/vegetated marsh ratios (UVVR) suggesting these metrics are broadly applicable indicators of microtidal marsh vulnerability. All sites are exhibiting a sediment deficit, with half the sites having projected lifespans of less than 350 years at current rates of sea-level rise and sediment availability. These results demonstrate that open-water conversion and sediment deficits are holistic and sensitive indicators of salt marsh vulnerability.

[1] U.S. Geological Survey, Woods Hole Coastal and Marine Science Center, 384 Woods Hole Road, Woods Hole, Massachusetts 02543, USA. [2] Virginia Institute of Marine Science, Department of Physical Sciences, 1375 Greate Road, Gloucester Point, Virginia 23062, USA. [3] Boston University, Department of Earth and Environment, 685 Commonwealth Avenue, Boston, Massachusetts 02215, USA. [4] University of Padua, Department of Civil, Environmental, and Architectural Engineering, Via 8 Febbraio, Padua 2-35122 Italy. Correspondence and requests for materials should be addressed to N.G. (email: nganju@usgs.gov).

Salt marshes provide critical ecosystem services but are perceived to be increasingly vulnerable to sea-level rise[1], sediment starvation[2], eutrophication[3] and limits on landward transgression[4]. One key element of marsh vulnerability is sediment availability: the geomorphic continuum of a salt marsh complex, consisting of tidal channels, intertidal flats and marsh plain, requires external sediment input to combat erosive forces such as waves and currents[5]. Sea-level rise poses an additional threat by modifying the vertical quasi-equilibrium of the entire system[1], increasing accommodation space and requiring more sediment for marsh survival. Prior work has demonstrated that salt marshes tend to maintain their vertical position through a combination of organic production[6] and inorganic sediment trapping[7]. However, salt marshes are rarely in horizontal equilibrium[8] and are constantly responding to competing constructive and destructive forces[5]. Physically, whether salt marshes advance or retreat is a function of external sediment supply from the watershed or marine end-members[5] as well as the possibility of landward transgression. Evaluating the mechanisms, magnitude and direction of sediment fluxes in marsh tidal channels provides an integrative metric for evaluating the trajectory of the salt marsh complex as a whole[9]. Microtidal marshes are the most vulnerable to future sea-level rise, given their location in the tidal frame and increased dependence on inorganic sediment supply relative to macrotidal systems[10,11].

Sediment fluxes in tidal systems are modulated by multiple time-varying processes. Tidal-timescale resuspension, advection and deposition can vary suspended-sediment concentration (SSC) and the ensuing fluxes by several orders of magnitude within tidal and spring-neap cycles[12,13]. Episodic riverine sediment delivery can elevate sediment concentration and fluxes over longer timescales due to increased input of watershed-derived sediment[14]. Variability of winds can drive differences in wave-induced sediment resuspension and therefore fluxes to and from tidal channels[15,16]. Because of these multiple timescales of sediment transport, tidal fluxes must be measured at high-temporal resolution ($<15$ min) with autonomous sensors over seasonal-to-annual timeframes to constrain variability and cover dominant conditions. Many landmark studies highlighted the importance and implications of sediment flux measurements[17,18], but few have applied the aforementioned modern techniques over a range of external forcings to reduce errors arising from infrequent temporal sampling[19].

Here we present a synthesis of marsh tidal channel flux studies from eight microtidal sites along the Atlantic and Pacific coasts of the United States (Fig. 1; Table 1). These sites span a spectrum of vegetation type, climatic forcing, tidal range, geomorphic setting and watershed land use. Each previously published study, and the studies first described in this paper, acquired continuous time-series of water flux and turbidity (calibrated to SSC with in-situ water samples) for at least 2 months, capturing a range of tidal (at least 100 tidal cycles) and episodic forcing. All but two studies followed U.S. Geological Survey protocols[20,21] for the determination of water and sediment flux in tidal channels. Net sediment fluxes were scaled to 1 year and normalized by total drainage area landward of the channel to allow for a spatially integrated, sediment-based assessment of marsh vulnerability. The net sediment budget is the sum of this measured flux and the sediment required to offset sea-level rise. We show that the sediment budgets of all sites scale with the unvegetated-vegetated marsh ratio (UVVR) and that the sediment-based lifespan of these marshes can also be predicted via the UVVR. We also identify a strong relationship between the flood-ebb SSC differential and the sediment budget, allowing for simplification of sediment transport monitoring. We find that

these sediment transport metrics and the UVVR are useful indicators of marsh trajectory, allowing for widespread mapping of marsh vulnerability using in-situ measurements and remote sensing.

## Results

**Sediment budgets scale with UVVR.** The sediment budgets of the marsh complexes consistently scale with the UVVR, an independent measure of marsh health (Fig. 2a). Root zone collapse[22], salinity dieback[23], herbivory[24] and waves[25] are mechanisms, among others, that convert vegetated marsh to open water. Despite this complexity, conversion to open water is ultimately a geomorphic process that modifies the sediment budget. The loss of vegetated marsh, both on the plain and fringes, reduces the long-term net trapping potential of a marsh complex by exposing the substrate and decreasing the momentum extraction and wave attenuation by stems[26]. In microtidal, sediment-starved settings, increased open water ultimately favours marsh erosion, which in turn increases the unvegetated area, in a positive feedback that deteriorates the entire marsh[27]. In certain cases, ponding on the marsh plain can result in permanent marsh loss through slumping and soil creep, with runaway erosion preventing recovery[27]. Sediment deposition is mostly controlled by vegetative influence on settling and direct capture, and therefore scales with the area of the vegetated surface. Erosive processes act on channel banks, marsh boundaries and within marsh ponds, and are therefore regulated by the area of the unvegetated surface. The correlation between UVVR and sediment budget is not necessarily causal; it is a convergence of two integrative indicators that bundle unseen processes across the landscape, which are difficult to measure independently. Though the marshes studied here occupy a wide parameter space (Table 1), the relationship between UVVR and sediment budget applies across all sites.

All of the complexes currently exhibit a sediment deficit once sea-level rise is accounted for (Table 1; Fig. 2a). The Fishing Bay site, adjacent to Chesapeake Bay, demonstrates the smallest deficit due to sizeable sediment import. The sediment import is due to an external source from the seaward direction, as well as consistent tidal mobilization processes[9]. The preponderance of sites with a sediment deficit is coherent with a documented reduction of external sediment sources and coinciding elevated rates of sea-level rise[2]. Contrastingly, accretion and elevation data[4,28] imply vertical stability though the integrative budget reveals three-dimensional instability. Idealized numerical modelling of the biogeomorphic patterns of vegetated marsh plains in the Venice Lagoon[29] shows that UVVR increases with decreasing sediment supply and increasing rates of sea-level rise, with stability at UVVR $\sim 0.13$ (Supplementary Fig. 1). This value is consistent with the UVVR value determined for the actual tidal watershed within the Venice Lagoon. Both observations and models indicate the existence of a biogeomorphic tipping point in the capability of a microtidal marsh complex to counter changes in external forcing.

**Sediment budget-based lifespan scales with the UVVR.** The UVVR also scales with the sediment budget-based lifespan of the marsh complex (Fig. 2b). This lifespan illustrates the time remaining until the equivalent sediment stored in the marsh plain above mean sea level is expended through a net sediment deficit. Though it is not an estimate of marsh plain extinction, it does indicate when the system will conceptually require lateral erosion to maintain sediment export needs and relative elevation of the entire system planform. The most vulnerable marshes in this study have lifespans between

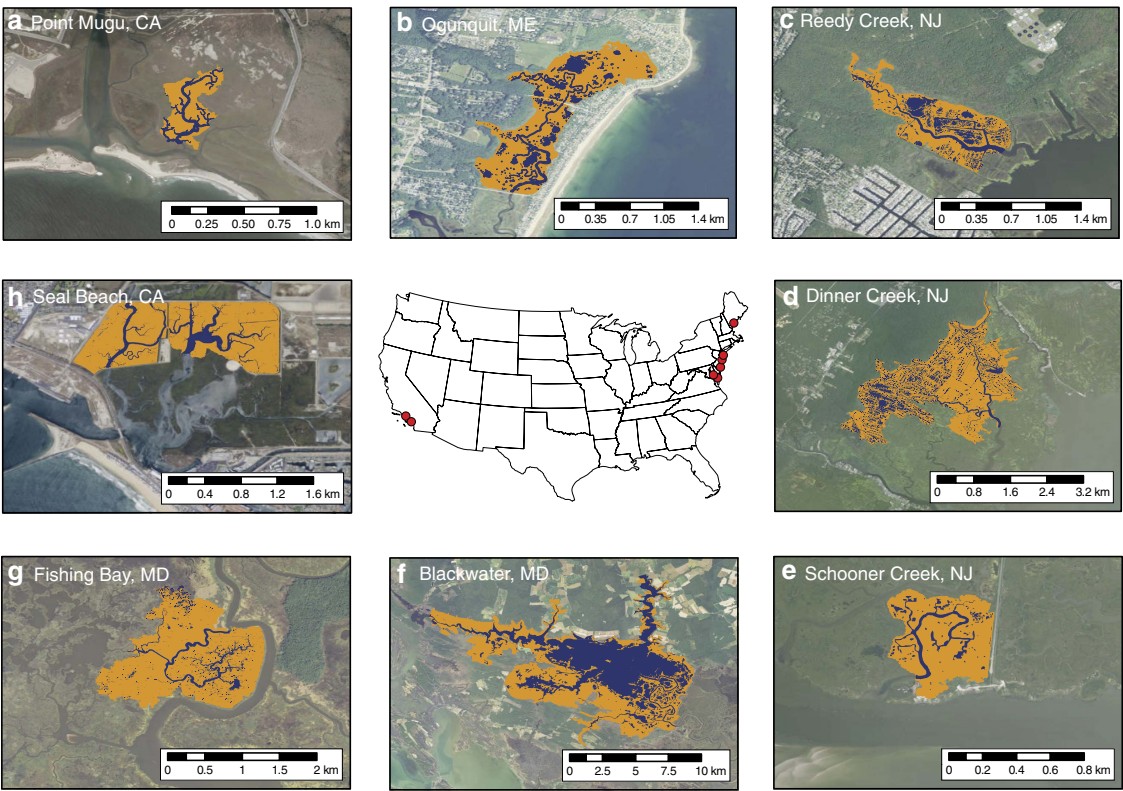

**Figure 1 | Aerial imagery and drainage delineation of eight salt marsh complexes.** (**a**) Pt. Mugu, California; (**b**) Ogunquit, Maine; (**c**) Reedy Creek, New Jersey; (**d**) Dinner Creek, New Jersey; (**e**) Schooner Creek, New Jersey; (**f**) Blackwater, Maryland; (**g**) Fishing Bay, Maryland and (**h**) Seal Beach, California. Unvegetated areas shown in blue, vegetated areas in orange. Drainage areas were established through the Hydrology toolbox in ArcGIS, for areas landward of sediment flux measurement location within the tidal channel; open water and marsh delineation was determined through aerial imagery and the National Wetlands Inventory. All images are oriented with north upwards. Imagery from the ArcGIS World Imagery Basemap.

### Table 1 | Sediment flux sites and associated data.

| Parameter/site | BW | FB | SB | PM | RC | DC | OG | SC |
|---|---|---|---|---|---|---|---|---|
| Bioclimate[51] | UMH | UMH | LTS | LTD | UMH | UMH | LSH | UMH |
| Marsh species | Spp, Sa[28] | Spp, Sa[28] | Spf, Sap[52] | Sap, Ds[52] | Spa[53] | Spa[53] | Spp[54] | Spa[53] |
| Surface lithology[51] | AC | AC | AC | AC | AC | AC | GLS | AC |
| Maximum tide range (m) | 0.35[28] | 0.75[28] | 2.5[37] | 1.6[37] | 0.31[38] | 0.73[38] | 2.1[39] | 1.0[19] |
| Vegetated area (km$^2$) | 36 | 1.6 | 1.2 | 0.12 | 0.37 | 3.6 | 1.2 | 0.18 |
| Unvegetated area (km$^2$) | 34 | 0.14 | 0.20 | 0.015 | 0.15 | 0.56 | 0.17 | 0.031 |
| UVVR | 0.94 | 0.090 | 0.17 | 0.12 | 0.40 | 0.16 | 0.14 | 0.17 |
| Elevation relative to MSL (m) | 0.24 | 0.38 | 0.71 | 0.72 | 0.29 | 0.43 | 1.34 | 0.53 |
| Mean SSC (mg l$^{-1}$) | 63[28] | 39[28] | 15[37] | 15[37] | 9.5[38] | 15[38] | 3.7[39] | 17[19] |
| Flood-ebb SSC differential (mg l$^{-1}$) | −17[28] | 5.1[28] | −1.1[37] | 2.6[37] | −0.78[38] | 1.7[38] | 0.55[39] | 1.9[19] |
| Sediment flux per unit area (kg m$^{-2}$ y$^{-1}$) | −0.46[9] | 0.56[9] | 0.061[52] | 0.25[52] | 0.020 | 0.15 | −0.025 | 0.21[19] |
| Sea-level rise[43] (m y$^{-1}$) | 0.0037 | 0.0037 | 0.0016 | 0.0023 | 0.0041 | 0.0041 | 0.0018 | 0.0041 |
| Net sediment budget (kg m$^{-2}$ y$^{-1}$) | −1.06 | −0.027 | −0.19 | −0.12 | −0.63 | −0.49 | −0.30 | −0.44 |
| Lifespan (y) | 83 ± 48 | 5,230 ± 3,000 | 1,430 ± 820 | 2,330 ± 1,340 | 170 ± 98 | 330 ± 190 | 1,640 ± 940 | 350 ± 200 |

AC, alluvium and coastal zone sediment; BW, Blackwater, MD; DC, Dinner Creek, NJ; Ds, *Distichlis spicata*; FB, Fishing Bay, MD; GLS, glacial lake sediment; LSH, lower supratemperate humid; LTD, lower thermomediterranean dry; LTS, lower thermomediterranean semiarid; OG, Ogunquit, ME; PM, Pt. Mugu, CA; RC, Reedy Creek, NJ; Sa, *Schoenoplectus americanus*; Sap, *Sarcocornia pacifica*; SB, Seal Beach, CA; SC, Schooner Creek, NJ. Bioclimate, marsh and lithology; Spa, *Spartina alterniflora*; Spf, *Spartina foliosa*; Spp, *Spartina patens*; UMH, upper mesotemperate humid.

83–350 years, in agreement with prior studies based on lateral erosion rates[8]. The similarity between the lifespan computed here through sediment fluxes and the lifespan estimated from erosion rates indicates that open-water conversion is the main destructive process in marshes, controlling the long-term fate of the marsh complex. Future changes in the rate of open-water conversion, sea-level rise and climatic factors may modulate sediment export and alter the lifespan prediction. As marshes lower in the tidal frame due to sea-level rise, there may be an increased propensity to trap more sediment and partially offset a sediment deficit[4]. From a whole-system perspective however, this depends on the reliability of an external sediment source, otherwise the likely source of trapped material is marsh edge erosion (that is, a cannibalization process[30]).

**Identification of sediment transport-based metrics.** Sediment flux (measured flux in absence of sea-level rise deficit) does not scale with temporally averaged SSC across all sites, but

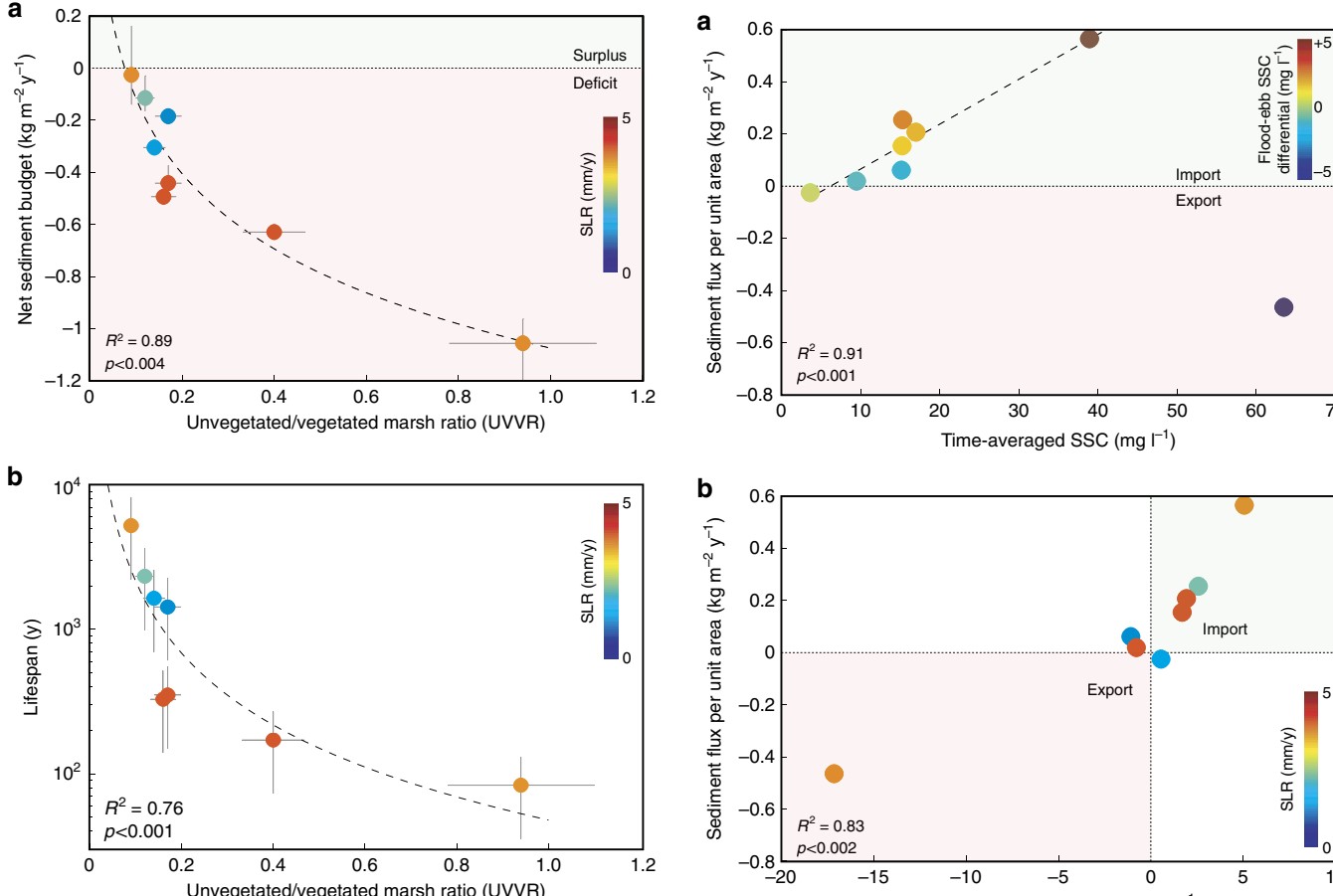

**Figure 2 | Relationships between sediment transport and geomorphic metrics.** (**a**) Ratio of unvegetated to vegetated marsh (UVVR) within the tidal channel drainage area versus net sediment budget (sediment flux minus supply needed to offset SLR) and (**b**) UVVR versus sediment-based lifespan of the marsh complex. Individual points are coloured to represent local sea-level rise. Regression statistics apply to fitted curves to data: (**a**) $y = -0.42 \ln x - 1.07$; and (**b**) $y = 48x^{-1.66}$. Error bounds correspond to potential uncertainty in (**a**) drainage area and UVVR; and (**b**) bulk density and UVVR. Note that sites with nearly zero sediment flux exhibit smaller errors in net sediment budget (**a**) due to a greater influence of the sea-level rise term.

**Figure 3 | Relationships between sediment transport metrics.** (**a**) Time-averaged suspended-sediment concentration (SSC) versus sediment flux per unit area, with points coloured by flood-ebb SSC differential and (**b**) flood-ebb SSC differential versus sediment flux per unit area from eight tidal marsh channels coloured by local sea-level rise. SSC is a reliable predictor of sediment flux for neutral-to-importing channels, but high SSC from marsh deterioration can result in large sediment export. The differential between flood and ebb SSC is a more reliable predictor of the integrative sediment budget. Regression statistics apply to fitted curves to data: (**a**) $y = 0.017x - 0.11$; site with large negative differential (BW) excluded; and (**b**) $y = 0.039x + 0.13$.

scales linearly across sites with a net or nearly zero import (Fig. 3a). The correlation between flux and SSC arises from the dependency of two flux components (advective and Stokes drift[13]) on mean SSC; a third component (dispersive flux) is dependent on temporal fluctuations in SSC (flood-ebb fluctuations). The absolute value of the flood-ebb SSC differential scales with mean SSC across all sites ($R^2 = 0.90$), indicating that the dispersive flux is correlated with mean SSC as well. This relationship is striking given differences in tidal range and advection mechanisms, but is expected given nonlinearities in fine sediment deposition and erosion with increasing SSC (refs 31,32).

The outlier to a universal relationship between SSC and sediment flux is the Blackwater complex, which has experienced decades of conversion to open water[33]. Elevated SSC at this site is a result of sediment liberation due to marsh deterioration[28], leading to elevated sediment export[9]. This disconnect highlights the fallacy of using a static SSC value to drive marsh vulnerability models[34]: the source of suspended sediment is more critical than the magnitude of SSC alone.

High SSC from internal sources such as lateral erosion can enhance vertical accretion on vegetated marsh[28], but will yield a false impression of stability from a three-dimensional perspective. The robust relationship between SSC and sediment flux for importing marshes does indicate that if an external sediment source and transport direction are identified, mean SSC may be an adequate predictor of sediment-based stability. However, the flood-ebb SSC differential[28] is a better predictor of sediment flux across all sites (Fig. 3b). This metric implicitly assigns directionality to SSC, and can diagnose trajectory as well as sediment source. Prior work also indicates that tidal systems approach morphodynamic equilibrium when flood-ebb SSC differentials and residual sediment approach zero[35].

## Discussion

Salt marsh vulnerability assessments often focus on point measurements of elevation, accretion, substrate quality and/or local sea-level rise to gauge absolute and relative vulnerability[4]. These methods provide limited insight due to a propensity

to capture vertical stability[4], underestimate spatiotemporal variability and neglect three-dimensional processes such as open-water conversion. The geomorphic interaction between tidal processes, waves, sea-level rise, sediment transport and the marsh complex are less frequently considered but illustrate a more holistic, integrated perspective on vulnerability. A healthy marsh complex must import sediment to counter waves, tidal currents and sea-level rise, while an unhealthy marsh is unable to resist these destructive forces and subsequently exports sediment[5]. The UVVR is a suitable indicator of marsh health, and supports this interpretation through its relationship with the marsh sediment budget. The relationship between sediment budget and UVVR presented here is direct observational evidence of the link between open-water conversion processes and sediment transport, and holds across a geomorphic and climatic spectrum of channelized, microtidal marshes. The conversion of salt marshes to open water is a gradual process that is typically observed over decadal timescales[36], but it is now possible to infer the sediment-based trajectory of a microtidal marsh complex based on a single, static image. Using present-day UVVR and tracking future changes to estimate the sediment budget and lifespan allows for widespread mapping of whole-system and spatially variable vulnerability across microtidal marshes worldwide.

## Methods

**Sediment transport metrics.** Total sediment fluxes ($Q_t$) from prior studies were scaled to yield yearly fluxes, and normalized by total marsh complex area ($A_t = A_v + A_{uv}$, where $A_v$ and $A_{uv}$ are vegetated and unvegetated area components of a pre-defined complex, respectively) landward of the flux measurement site (see below), yielding an annual sediment flux per unit area:

$$Q_s = Q_t A_t^{-1} \qquad (1)$$

Positive values of $Q_s$ indicate sediment import. For the previously unpublished sites (Ogunquit, Maine; Dinner Creek, New Jersey; and Reedy Creek, New Jersey), sediment fluxes were measured following standard methods as established by the prior studies[9]. Briefly, autonomous sensors measuring vertical profiles velocity and near-bottom turbidity were deployed on the channel seabed. Periodic surveys of cross-sectional velocity distribution, channel area and sediment concentration were used to convert these proxy values to cross-sectionally representative values[9]. Turbidity was converted to SSC using collected water samples at individual sites, following standard methods. The basic time-series data of velocity and turbidity for these sites have been quality-controlled and published[37–39]. Mean SSC and flood-ebb SSC differential were calculated following prior methods[28]. Deployments captured over 100 tidal cycles, and several high water events from both extreme spring tides as well as frontal passages (Supplementary Table 1). Additionally, most of the wetland complexes were constrained by topographic controls including uplands and hardened structures. The two sites with the largest tide range were constrained on all edges by roads and uplands (sites SB and OG, respectively). This reduces the uncertainty involved with unchannelized flow at the highest tides. There may still be significant uncertainty in the extrapolation of sediment flux estimates given annual and episodic variability. Individual storms with >1 year recurrence intervals can deposit substantial quantities of sediment on the marsh plain and alter long-term trajectory[40].

The net sediment budget (that is, deficit or surplus; $Q_b$) was calculated as:

$$Q_b = Q_s - \rho_{min} \times SLR_{local} \qquad (2)$$

where $\rho_{min}$ is a representative minimum dry bulk density (kg m$^{-3}$) and $SLR_{local}$ is the local rate of sea-level rise (m y$^{-1}$), including vertical land motion. Positive values of $Q_b$ indicate a sediment surplus. Using a comprehensive compilation of density data[41], we computed the mean and standard deviation of all brackish and salt marsh sites, which yielded a density of 373 ± 214 kg m$^{-3}$ (that is, average minimum density of 159 kg m$^{-3}$). For the budget calculation, we use the minimum value across all sites due to high spatial variability in surficial sediment properties even within individual complexes[42], and to generate a conservative estimate of marsh survival given uncertainties over future autochthonous deposition (which would tend to yield a low density). In this case, the density is required to convert the sea-level rise rate to a representative potential accretion. The local rate of sea-level rise over the entire period of record was taken from the nearest sea-level trend location[43], except for Point Mugu and Seal Beach, California, which were each located approximately equidistant from two trend locations; the values were averaged.

The sediment budget-based lifespan ($L_{sed}$) was calculated as:

$$L_{sed} = E_m \times \rho_{mean} Q_b^{-1} \qquad (3)$$

where $E_m$ is the mean elevation of the vegetated marsh plain relative to local mean sea-level (see below). $L_{sed}$ represents the time needed to export the entire volume of marsh above mean sea level ($\rho_{mean} E_m$) with a sediment discharge equal to $Q_b$, and $\rho_{mean}$ is a representative mean dry bulk density. We use the mean value in this case to represent the density contained within the remaining marsh plain, as opposed to a future potential deposition used in the sediment deficit calculation. This metric is based on the observation that microtidal salt marshes have an elevation above mean sea level, while tidal flats are below mean sea level[44]. Therefore loss of the marsh volume above mean sea level triggers the transition to tidal flat. Scaling errors in the sediment flux calculation (for example, using several months of data extrapolated to decadal timescales) would cascade to this lifespan calculation.

**Geomorphic metrics.** Elevation referenced to NAVD88 was calculated using the USGS National Elevation Dataset (NED; http://nationalmap.gov/elevation.html), and converted to local mean sea level using NOAA's VDatum software (http://vdatum.noaa.gov/docs/userguide.html). The boundaries for each study area were identified by the surface slope that drains the marsh water through the sediment flux measurement cross-section, following prior studies[45,46]. Hydrology tools in ArcMap Spatial Analysis toolbox were used to identify these hydrologic units based on the ~3-m resolution NED. Each hydrological unit was further clipped to the saltmarsh boundaries based on the wetland classification maps from the U.S. Fish & Wildlife Service's National Wetland Inventory (NWI; http://www.fws.gov/wetlands). For smaller study areas, where the NWI layer was not detailed enough to resolve the change in vegetation cover as seen in the high resolution imagery, a maximum elevation based on the great diurnal range of tides (GT) as obtained from VDatum was applied to clip the hydrologic unit boundary.

Unvegetated (that is, ponds, channels, flats) and vegetated areas within each marsh complex boundary were determined using the infrared band from the 1-m resolution National Agriculture Imagery Program (NAIP; http://www.fsa.usda.gov/programs-and-services/aerial-photography/imagery-programs/naip-imagery) and ArcGIS World Imagery (WI) basemap. NAIP images were analysed with a combination of isodata clustering[47] and maximum likelihood analysis[48] to categorize each pixel into a preset number of classes. The higher resolution WI images were overlaid with these classes to determine which classes constituted water and vegetation. The range of UVVRs ($A_{uv}/A_v$) between the eight sites allows for a time-space substitution, with the implication that sediment export increases as a marsh converts to open water, and UVVR increases. We chose to use a static, modern estimate of UVVR instead of attempting a change in UVVR due to the difficulty in obtaining consistent imagery between all sites over multiple years, and detecting changes over short timescales. Use of a modern UVVR also allows for modern assessment of sediment budget and/or marsh trajectory in response to sea-level rise.

We constrained errors associated with the methodology by using modern LiDAR bare-earth topography[49] at applicable sites in New Jersey to calculate elevation, drainage areas and UVVR. The NGS data set shares the same vertical datum, North American Vertical Datum of 1988 (NAVD88), with the NED data set but has a higher resolution (<1 m$^2$). We also replaced the 1-m resolution NAIP images used for delineating the vegetated/unvegetated areas in each marsh complex with the Edwin B. Forsythe National Wildlife Refuge (EBF-NWR, New Jersey) marsh delineation map. The EBF-NWR consists of polygons shapes depicting the external boundaries of lands and waters that are approved for acquisition by the U.S. Fish and Wildlife Service (USFWS). The same procedure as the original method was followed to process the data; however, because the vegetated and unvegetated data were acquired from the EBF map the isodata clustering and maximum likelihood analyses were not required. First, the LiDAR topography was gridded to a 0.5-m resolution raster map for watershed analyses with the Hydrology tools in ArcMap Spatial Analysis toolbox. The boundaries were then clipped to the vegetated and unvegetated areas according to the EBF-NWR delineation.

The change in the calculated vegetated and unvegetated surface area in Reedy Creek was +5% and −11%, respectively. This translated to a reduction in UVVR from 0.40 to 0.34. The change in mean marsh elevation was 1 cm. In the case of Dinner Creek vegetated and unvegetated areas were modified by −27% and −13%, respectively, resulting in an increase in UVVR from 0.16 to 0.19. The change in mean marsh elevation was 3 cm. The change in overall area was greatest at Dinner Creek (−25%), due to LiDAR artefacts incorrectly blocking drainage pathways. Therefore we consider maximum error in the area calculation as 25%, maximum error in UVVR as 17% and maximum error in elevation as 5.5%.

**Error propagation to correlations.** To assess the influence of errors on the correlations presented in Fig. 2, we simulated 10,000 realizations with maximum random errors of 17% and 25% for UVVR and area (affecting net sediment budget) for the UVVR-sediment budget correlation, and maximum random errors of 214 kg m$^{-3}$ (see above) for bulk density and 17% for UVVR for the UVVR-lifespan correlation. Coefficients of determination were computed for each

realization using the same fitting parameters presented in Fig. 2. Median $r^2$ values were 0.88 for the UVVR-sediment budget correlation (Supplementary Fig. 2A) and 0.82 for the UVVR-lifespan correlation (Supplementary Fig. 2B). Minimum values were 0.72 and 0.43 respectively, indicating that the correlations are robust despite potential errors.

**Idealized biogeomorphic modelling.** We used a biogeomorphic model of salt-marsh evolution[29] to address the effects of sediment supply and rates of sea-level rise on the UVVR. The model uses a simplified treatment of the two-dimensional shallow water equations[50] to describe marsh hydrodynamics and determines platform evolution by coupling the Exner sediment balance equation with the solution of an advection-dispersion equation over the marsh platform. Changes in marsh surface elevation are dictated by the balance between erosion, inorganic deposition through settling and particle capture by vegetation, and organic soil production deriving from the competition among different vegetation species. Numerical experiments were carried out for the tidal watershed of an actual channel network within the San Felice salt marsh, in the Venice Lagoon, Italy. Constant SSCs of 10, 20 and $30 \, mg \, l^{-1}$ (representing external sediment supply) were specified within the channel network, with imposed sea-level rise rates of 3, 5, and $7 \, mm \, y^{-1}$. Net sediment deficit was calculated in the following equations (1) and (2). Different sediment supplies and rates of sea-level rise promoted the formation of different marsh topographic configurations, characterized by the transition of vegetated marsh portions to unvegetated areas, thus influencing the UVVR ratio.

**Data availability.** All time-series data, except for the Schooner Creek site[19], can be accessed at the USGS Oceanographic Time-Series Database at http://stellwagen.er.usgs.gov/.

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

## Acknowledgements

This study was part of the Estuarine Physical Response to Storms project (GS2-2D), supported by the Department of the Interior Hurricane Sandy Recovery program and the U.S. Geological Survey Coastal and Marine Geology Program. M.L.K. acknowledges funding from NSF GLD 1529245, NSF Coastal SEES 1426981, NSF LTER 1237733, and the USGS Climate and Land Use Change Research and Development Program. S.F. acknowledges funding from NSF LTER 1237733, NSF LTER 1637630 and the Department of the Interior Hurricane Sandy Recovery program (project GS2-2D). Marinna Martini, Ellyn Montgomery, Jonathan Borden, Christine Sabens, Patrick Dickhudt, Sandra Brosnahan, Steven Suttles, Roland Hagan, Karen Thorne, Chase Freeman, Christopher Sherwood, Dan Nowacki, Patrick Brennand and Kyle Derby are acknowledged for assistance with this study. Namsoo Suk provided data for site SC. Erika Lentz and two anonymous reviewers provided constructive comments on the manuscript.

## Author contributions

N.K.G., Z.D. and M.L.K. designed the study; Z.D. performed GIS analyses; N.K.G. performed flux calculations; A.D'A. and L.C. assisted in developing the UVVR versus sediment budget concept; S.F. developed the lifespan concept; all authors contributed to the drafting of the manuscript.

## Additional information

**Competing financial interests**: The authors declare no competing financial interests.

**How to cite this article**: Ganju, N. K. *et al.* Spatially integrative metrics reveal hidden vulnerability of microtidal salt marshes. *Nat. Commun.* **8**, 14156 doi: 10.1038/ncomms14156 (2017).

**Publisher's note**: 

