## [Peer Review File · Nature Communications]

Reviewers' comments:

Reviewer #1 (Remarks to the Author):

The manuscript "Spatially integrative metrics reveal hidden vulnerability of salt marshes" elaborates on the importance of spatially integrated rather than point metrics to assess salt marsh vulnerability. Via a comparison between integrated sediment budgets of 8 salt marshes and their unvegetated/vegetated ratio a link between an easy accessible metric and salt marsh vulnerability is constructed.

The main conclusion of this study is that not only vertical accretion, but also two-dimensional, effects need to be considered to evaluate whether wetlands will persist in the face of global change. This is an important progression in evaluating salt marsh vulnerability. Therefore this study is a major contribution for our understanding of which metrics should be monitored when evaluating wetland resilience.

The used approach is a valid and supported by the clearness of the presented data. Previous work is referenced appropriately.

I think the manuscript is well written and should be accepted with minor revisions. My main criticism consists in the method of delineating watersheds, which are in the manuscript referred to as "salt marsh complexes". The utilization of Hydrology toolbox in ArcGIS to delineate drainage areas is problematic in flat areas, especially when vegetation is present influencing the elevation signal (which depends on how they are accounted for in the NED dataset).

Since this delineation has major impacts on the results and further the usability of the UVVR metric for salt marsh vulnerability I would suggest adding an additional explanation and show the robustness of the method.

Line 57: I would suggest renaming the y-axis figure 2a to increase clarity

Line 83: As mentioned above how dependent are the shown ratio for stability on delineation of drainage areas

Line 99: Here the calculation for sediment deficit or surplus is discussed, and explained in the methods (line 283). Why is the assumed uniform density chosen that low (150 kg/m³)?

Line 106: replace tide by tidal

Line 115: "flood-ebb SSC a better predictor of sediment flux across all sites", I think the authors miss a chance here to utilize the flood-ebb SSC as a proxy for the morphodynamic equilibrium of the systems, during which following Lanzoni 2012 long-term residual transport should be cancelled each other out

Line 127: needs a reference to be added

Line 134, 269, 297-300: see comment Line 83 → to make this claim delineation methods need to be further explained

Line 275: maybe a short explanation on the measurements would be useful

Line 283: Does local sea level include subsidence?

Reviewer #2 (Remarks to the Author):

A. Summary of the Key Results

This is an interesting paper that presents an analysis of a series of salt marsh systems' sediment budgets based on between 8 and 10 months (April - November, August-January, March-December) of turbidity measurements at individual marsh inlet sites (references 32, 34, 35). The authors then convert the observed net in- or out-flux of sediment to a long-term average annual flux by extrapolation using the average elevation of the marsh surfaces as obtained from USGS national elevation data and local tide-gauge station data (to derive net in- and out-flux for all tides that flood the marsh surface over the period of a year). Using information on sediment bulk density (the relative weight per volume) of marsh sediments alongside estimates of total marsh area flooded through the respective inlet, they convert the net in- or out-flux of sediment into elevation gain or loss of the marsh surface. This allows the authors to compare the estimated expected elevation gain or loss directly to sea level rise rates and thus predict the time until marsh elevation falls below mean sea level, which the authors take to be the critical point at which the marsh cannot re-establish itself. The authors then plot this predicted net sediment surplus/deficit and predicted life-span against the ratio of unvegetated to vegetated marsh surface area derived from the static imagery used to construct the NWI wetlands map and find a negative correlation, suggesting that unvegetated to vegetated marsh surface area ratio can be used as an indicator of the marsh's stability / life span.

B. Originality and interest

In the context of a search for easily applicable universal predictors for coastal marsh survival, this analysis and conclusion is very appealing. It would be a great advancement to be able to communicate to anyone responsible for coastal marsh conservation and management that simple metrics can be easily applied to derive information about any marsh system's future evolution. Marsh systems are complex biologically mediated landforms which respond to a multitude of external and internal processes that affect the relationship between sediment fluxes and marsh surface elevation change. Thus this type of approach is challenging (although perhaps too much so, considering my points below) and novel.

C. Data & methodology: validity of approach, quality of data, quality of presentation

The method, results, and interpretation, contain a series of flaws. The following points are the main reasons why I would urge the authors to review (expand, and refine) their research prior to publication.

1. Perhaps the biggest single problem with this study is the fact that it uses a static metric (vegetated versus unvegetated marsh area (UVVR)) as a key variable for the prediction of future marsh sedimentary status by way of time-space substitution (i.e. using the static metric for a range of marsh systems, correlating this against measured net sediment flux, and basing the argument of the future prediction of an individual marsh's life span from UVVR on the outcome of this correlation). Using this method, the authors thus assume that the UVVR represents a direct response to the net sediment flux as measured at the sites during a specific 8-10 month period of a particular year. This seems to me a fundamentally flawed argument, as (a) UVVR is the result of many years of vegetation growth/decay on the marsh surfaces (and there are likely to be significant lags in the system between the processes that drive a change in UVVR and the resulting UVVR itself), and (b) it is known that net sediment fluxes may contain significant inter-annual variability (i.e. an individual 8-10 months of turbidity measurements as used here cannot be used as an integrative measure of sediment budgets over the time scales over which a particular vegetation cover ratio (UVVR) results; this is certainly the case for macrotidal systems (see point 2 below)).

2. The second most important problem with this study is the assumption that surface elevation change of the marsh complex can be calculated simply through assuming (a) that the bulk density of existing, deposited sediments, is the same as that of the newly deposited sediments measured through the turbidity sensors used in this study to derive sediment fluxes, and (b) uniform bulk density of the sediment that is imported or exported over the annual time-scale. Compaction of sediments has been shown to be highly variable and dependent on a multitude of organic and climatological processes, all of which operate on time-scales similar to those of sea level rise (see e.g. Allen, 2000; Cahoon et al., 2006; Day and Giosan, 2008). The authors themselves admit that there is 'high spatial variability in surficial sediment properties even within individual marsh complexes' (line 285) - such properties can also be expected to vary markedly with time and particularly with climatic changes that will accompany sea level rise over commensurate time scales (i.e. changes in atmospheric temperature and precipitation that can significantly alter the degree of organic and inorganic matter production as well as the release of minerogenic sediment from offshore and alongshore sources due to altered wave exposure of eroding shores and the sea bed....

D. Appropriate use of statistics and treatment of uncertainties

In addition to the concerns expressed above around the extrapolation of suspended particular matter measurements from 8-10 month to average annual rates, there are other particular instances where further detail and error/uncertainty assessment is required:

Line 297: The vertical accuracy of this data is, at best, 8 cm. This raises concerns around the accuracy of the estimates of elevation relative to mean sea level, as this is used to derive 'life span' based on the computed net sediment flux and marsh area.

Line 301: The authors mention the horizontal resolution but there is no mention of vertical accuracy or resolution, both are critical to the estimation of 'life span' (see above)

Line 306: It would be useful to know what the potential error is that is introduced by this adjustment.

The authors state (line 298-301) that "The boundaries for each study area were identified by the surface slope that drains the marsh water through the sediment flux measurement cross-section. Hydrology tools in ArcMap Spatial Analysis toolbox were used to identify these hydrologic units based on the ~3-m resolution NED." Given the sensitivity of marsh surface flows, in particular macro-tidal systems, to very small (< 10 cm) differences in elevation and to wind conditions during the time of inundation, the methods applied in this paper are, at best, only applicable to very clearly topographically constrained, micro-tidal, marsh systems. It is not clear from the text whether the marsh systems used in this paper are topographically constrained in this way or whether tidal exchange on large spring tides could lead to significant errors in the sediment budgets computed from the single point turbidity sensor data, as has been suggested by thorough studies of sediment budgets in macro-tidal systems nearly 20 years ago (e.g. Murray and Spencer, 1997).

E. Conclusions: robustness, validity, reliability

The key issues around the validity of the approach of the study have been outlined under 'D' above. In addition, the following issues somewhat reduce the robustness of the conclusions:

1. The tidal range within the marsh systems used within this analysis is dominated by micro-tidal systems (6 out of the 8 sites used), with the other two being barely meso-tidal systems (maximum tidal range 2.1 to 2.5 m). If all sites were classified based on mean tidal range, all

would classify as micro-tidal. At the very least, the conclusions must thus be stated as being limited to micro-tidal marsh systems only. There are in fact many reasons why the results of this analysis can be expected to be vastly different for meso- to macro-tidal systems, e.g.:

a. Meteorological conditions play an important role in controlling tidal fluxes of sediment (see e.g. Manning and Bas, 2006; Uncles and Stephens, 2010), and during over-marsh water depths, flow above the vegetation canopy can be significantly correlated with decadal-scale climatic oscillations affecting sea level fluctuations (e.g. Philips and Crisp, 2010; Philips et al, 2013).

b. Tidal flat - salt marsh transitions in meso- to macro-tidal systems, particularly on open coast settings, are often much more dynamic and responsive to variations in meteorological conditions and wave approach angles and energies; including the possibility for marsh edge erosion resulting from the occurrence of individual extreme events. Such events can conceivably cause preferential tidal flat lowering and cliff formation with a runaway effect through reflected wave energy at the cliff causing marsh edge retreat, rather than any alteration in sediment supply or net sediment flux (e.g. Kirby and Kirby's 2008 work on the climatological controls of tidal flat elevations).

2. All marsh systems included in the analysis are North American and found within very limited biogeographical zones (see Figure 1); the response of such marshes to all external drivers mentioned in the paper as well as their internal biological dynamic is fundamentally different to systems elsewhere in the world (see e.g. the diverse and geomorphologically complex and varied systems of the North Atlantic region (see e.g. Allen, 2000), and those in Asia and in the sub-tropical regions of the world), as differences in vegetation composition alter the above- to below-ground biomass ratios, interaction with tidal and wave flows, sediment capture efficiency, not to mention their response to elevated CO₂ and altered atmospheric temperatures as a result of climatic changes that are to be expected over the sea-level rise time-scales considered here.

3. The study assumes that sea level rise as derived from past tide gauge data records can simply be extrapolated for the purpose of predicting marsh sedimentary status into the future. It is likely, however, that sea level rise rates are non-linear over time into the future.

F Suggested improvements: experiments, data for possible revision

The idea behind this paper is a very attractive and justifiable one, however, the data base used to approach it and the analysis is not (yet) scientifically robust and rigorous enough. I suggest the authors either:

(a) rewrite the paper as a rather more geographically restricted study, making clear that the conclusions are restricted to a particular type of marsh system that is not representative of salt marshes more generally/globally and addressing the non-linearity that is likely involved in marsh response to drivers such as sea-level rise and accompanying climatic and sediment supply changes over decadal time-scales; or

(b) broaden the study to include data from marsh systems with vastly different tidal ranges, sediment and vegetation types, and re-analyse those data with a view to establishing the broad relationships they are searching for.

However, given the complexity of salt marsh system functioning in a wide variety of climatological and sedimentological settings, the three-dimensional nature with which sediment is introduced and exported from marsh systems, and, furthermore, the expected complexity of marsh system response to non-linear sea level and external tidal sediment loading forcing over decadal time-scales, it would be surprising if the method that the authors are suggesting would yield sensible results, certainly at the larger scale implicit in option (b).

G References: appropriate credit to previous work?

There seems to be a tendency for over-reliance on the authors' own publications and on US literature. There is a wider body of literature out there that informs on the complexities that are

currently not addressed... (see below).

Allen, J. R. L. (2000). Morphodynamics of Holocene salt marshes: A review sketch from the Atlantic and Southern North Sea coasts of Europe. *Quaternary Science Reviews*, 19(12), 1155-1231. doi:10.1016/S0277-3791(99)00034-7

Cahoon, D. R., Hensel, P. F., Spencer, T., Reed, D. J., McKee, K. L., & Saintilan, N. (2006). Coastal Wetland Vulnerability to Relative Sea-Level Rise: Wetland Elevation Trends and Process Controls. In J. T. A. Verhoeven, B. Beltman, R. Bobbink, & D. F. Wigham (Eds.), *Wetlands and Natural Resource Management* (pp. 271-292). Springer.

Day, J. W., & Giosan, L. (2008). Survive or subside? *Nature Geoscience*, 1(March), 156-157.

Hay, C. C., Morrow, E., Kopp, R. E., & Mitrovica, J. X. (2015). Probabilistic reanalysis of twentieth-century sea-level rise. *Nature*, 517(7535), 481-484. doi:10.1038/nature14093

Kirby, J. R., & Kirby, R. (2008). Medium timescale stability of tidal mudflats in Bridgwater Bay, Bristol Channel, UK: Influence of tides, waves and climate. *Continental Shelf Research*, 28(19), 2615-2629. doi:10.1016/j.csr.2008.08.006

Manning, A. J., & Bass, S. J. (2006). Variability in cohesive sediment settling fluxes: Observations under different estuarine tidal conditions. *Marine Geology*, 235(1-4), 177-192. doi:10.1016/j.margeo.2006.10.013

Murray, A. L., & Spencer, T. (1997). On the wisdom of calculating annual material budgets in tidal wetlands. *Marine Ecology Progress Series*, 150(1-3), 207-216. doi:10.3354/meps150207

Phillips, M. R., & Crisp, S. (2010). Sea level trends and NAO influences: The Bristol Channel/Severn Estuary. *Global and Planetary Change*, 73(3), 211-218. doi:10.1016/j.gloplacha.2010.06.005

Phillips, M. R., Rees, E. F., & Thomas, T. (2013). Winds, sea levels and North Atlantic Oscillation (NAO) influences: An evaluation. *Global and Planetary Change*, 100, 145-152. doi:10.1016/j.gloplacha.2012.10.011

Spencer, T., Möller, I., Rupprecht, F., Bouma, T. J., van Wesenbeeck, B. K., Kudella, M., ... Schimmels, S. (2016). Salt marsh surface survives true-to-scale simulated storm surges. *Earth Surface Processes and Landforms*, 41(4), 543-552. doi:10.1002/esp.3867

Uncles, R. J., & Stephens, J. a. (2010). Turbidity and sediment transport in a muddy sub-estuary. *Estuarine, Coastal and Shelf Science*, 87(2), 213-224. doi:10.1016/j.ecss.2009.03.041

H Clarity and context: lucidity of abstract/summary, appropriateness of abstract, introduction and conclusions

The style in which the paper is written is clear and lucid overall with an appropriate explanatory abstract, general introduction and conclusions are written in a clear and concise style.

LINE BY LINE COMMENTS to support the above statements:

Line 4: vulnerability to what? (unclear sentence)

Line 9: this statement is not really true, as it is extremely critical whether the surplus arises from high vertical accretion and is accompanied by edge erosion...

Line 11: No, it can also be addressed through decadal scale annual accurate elevational surveys and tracking of morphological marsh and tidal flat change...

Line 12: This is rather geographically restricted and very different vegetation cover to, e.g., NW Europe (reference 5, with which this is contrasted, includes NW European marsh systems, so this comparison is not quite justified...)

Line 49: What vegetation types were covered? These seem very different to marshes elsewhere in the world...

Line 51: It is not clear at what spatial resolution the measurements were obtained. There are problems around point-sampling (see also Murray and Spencer (1997)) that are not addressed here.

Line 52: Again, Murray and Spencer (1997) show that this sort of 'scaling' is very likely to lead to significant sediment budget errors...

Line 55: It seems not accurate to say that the net sediment budget as measured in terms of tidal

fluxes is the sediment required to offset sea-level rise; the authors themselves state earlier that organically produced sediment can contribute significantly... such sediment is not captured with this approach here...

Line 61: the authors claim that 'loss of vegetated marsh [...] reduces the net trapping potential of a marsh complex by liberating sediment from the substrate'. This is highly unlikely, at least over short time-scales and is highly dependent on the exposure of the unvegetated marsh surface (i.e. where it is located relative to incident wave approach). Spencer et al. (2015) showed that even storm wave conditions failed to liberate sediment from surfaces from which above-ground biomass had been removed, as root and organic matter within the soil substrate prevented re-suspension of sediments. At the very least, this introduces significant inter-annual (and potentially decadal) time-lags before sediment can be mobilised...

Line 62: this alone, however, is not enough to drive a net loss in vegetated marsh surface - in some NW European systems, such spatially differentiated erosion leads to accretion elsewhere in the marsh and to the maintenance of the marsh sediment budget in spite of marsh loss at the marsh margins (e.g. discussion in Wolters et al. 2005). Where manmade landward barriers to marsh growth are removed, field evidence also suggests that rapid landward and upward growth of marshes can occur and more than balance seaward fringe erosion (e.g. Spencer et al. 2011):

Wolters, M., Bakker, J. P., Bertness, M. D., Jefferies, R. L., & Möller, I. (2005). Saltmarsh erosion and restoration in south-east England: Squeezing the evidence requires realignment. *Journal of Applied Ecology*, 42(5), 844-851. doi:10.1111/j.1365-2664.2005.01080.x

Spencer, T., Friess, D. A., Möller, I., Brown, S. L., Garbutt, R. A., & French, J. R. (2011). Surface elevation change in natural and re-created intertidal habitats, eastern England, UK, with particular reference to Freiston Shore. *Wetlands Ecology and Management*, 20(1), 9-33. doi:10.1007/s11273-011-9238-y

Line 64: but elsewhere (see previous comment), it is suggested that this process can easily be counter balanced by landward migration of the marsh, space permitting... Also, open water only favors marsh erosion by water currents where those currents are strong enough to erode the sediment - and, as tidal currents are low, that is only the case where waves are of a magnitude high enough to cause erosion - a situation that can only be generated if the individual patches of unvegetated marsh are large enough to allow wind to generate sufficiently high waves over sufficient fetch distances. The study presented here does not take account in any way of the dimension/distribution of unvegetated areas within the marsh complex. It is simply not true that 'open water ultimately favors marsh erosion' (line 63).

Line 66: yes, but this is all too frequently constrained by man-made structures on the landward margin; where such structures are not present, marsh expansion on the landward margins is made possible and can maintain overall marsh area even where seaward margin erosion is taking place

Line 114: ... but only if one assumes that the geometric context which determines the horizontal constraints to tidal flow around the marsh remains fixed over the time-scales over which predictions are made... within the 10-300 year time scale, that is likely to be a flawed assumption...

Line 294: This reference is based on micro-tidal marsh systems only.

Line 295: again, this applies to micro-tidal marsh systems only.

Line 297: The vertical accuracy of this data is, at best, 8 cm. This raises concerns around the accuracy of the estimates of elevation relative to mean sea level, as this is used to derive 'life span' based on the computed net sediment flux and marsh area.

Line 301: The authors mention the horizontal resolution but there is no mention of vertical

accuracy or resolution, both are critical to the estimation of 'life span' (see above)

Line 306: It would be useful to know what the potential error is that is introduced by this adjustment.

Reviewer #1 comments:

My main criticism consists in the method of delineating watersheds, which are in the manuscript referred to as "salt marsh complexes". The utilization of Hydrology toolbox in ArcGIS to delineate drainage areas is problematic in flat areas, especially when vegetation is present influencing the elevation signal (which depends on how they are accounted for in the NED dataset).

Since this delineation has major impacts on the results and further the usability of the UVVR metric for salt marsh vulnerability I would suggest adding an additional explanation and show the robustness of the method.

R1.1 We recognize this concern, and have used additional data to estimate the error of this procedure. We used a recent high-resolution, bare-earth LiDAR dataset of New Jersey at two sites (Reedy Creek and Dinner Creek, NJ) to recompute areas. Errors in the area calculation, UVVR, and elevation (25%, 17%, and 5.5%, respectively), and the errors cascaded to sediment flux and lifespan are now noted in Figure 2 and the Methods section. For bulk density, the lifespan calculation was modified to account for uncertainty in bulk density, following recommendation from both reviewers.

We have added these methods and implications on lines 228-249:

“We constrained errors associated with the methodology by using modern LiDAR bare-earth topography⁵³ at applicable sites in New Jersey to calculate elevation, drainage areas, and UVVR. The NGS dataset shares the same vertical datum, North American Vertical Datum of 1988 (NAVD88), with the NED dataset but has a higher resolution (<1 m²). We also replaced the 1-m resolution NAIP images used for delineating the vegetated/unvegetated areas in each marsh complex with the Edwin B. Forsythe National Wildlife Refuge (EBF-NWR, New Jersey) marsh delineation map. The EBF-NWR consists of polygons shapes depicting the external boundaries of lands and waters that are approved for acquisition by the U.S. Fish and Wildlife Service (USFWS). The same procedure as the original method was followed to process the data, however, because the vegetated and unvegetated data were acquired from the EBF map the isodata clustering and maximum likelihood analyses were not required. First, the LiDAR topography was gridded to a 0.5-m resolution raster map for watershed analyses with the Hydrology tools in ArcMap Spatial Analysis toolbox. The boundaries were then clipped to the vegetated and unvegetated areas according to the EBF-NWR delineation.”

“The change in the calculated vegetated and unvegetated surface area in Reedy Creek was +5% and -11%, respectively. This translated to a reduction in UVVR from 0.40 to 0.34. The change in mean marsh elevation was 1 cm. In the case of Dinner Creek vegetated and unvegetated areas were modified by -27% and -13%, respectively, resulting in an increase in UVVR from 0.16 to 0.19. The change in mean marsh elevation was 3 cm. The change in overall area was greatest at Dinner Creek (-25%), due to LiDAR artifacts incorrectly blocking drainage pathways. Therefore we consider maximum error in the area calculation as 25%, maximum error in UVVR as 17%, and maximum error in elevation as 5.5%.”

Additionally, we have simulated Figure 2 with 10000 realizations of UVVR vs. sediment budget, and UVVR vs. lifespan with these random errors imposed on the actual values (as maximum errors). The new supplementary figure shows that the distribution of r^2 values supports the robustness of these relationships, lines 251-259:

“To assess the influence of errors on the correlations presented in Figure 2, we simulated 10000 realizations with maximum random errors of 17% and 25% for UVVR and area (affecting net sediment budget) for the UVVR-sediment budget correlation, and maximum random errors of 214 kg/m³ (see above) for bulk density and 17% for UVVR for the UVVR-lifespan correlation. Coefficients of determination were computed for each realization using the same fitting parameters presented in Figure 2. Median r^2 values were 0.88 for the UVVR-sediment budget correlation (Supplementary Figure 2A) and 0.82 for the UVVR-lifespan correlation

(Supplementary Figure 2B). Minimum values were 0.72 and 0.43 respectively, indicating that the correlations are robust despite potential errors.”

We have also added error bars to Figure 2a and 2b to demonstrate the strength of these relationships even with additional uncertainty, and included citations of other studies that have used the ArcGIS watershed tool (Vandenbruwaene et al., 2012, 2013).

We have also clarified that most of our sites are bounded by uplands or artificial structures (Supplementary Table 1), and are therefore insensitive to delineation of boundaries using a DEM.

Line 57: I would suggest renaming the y-axis figure 2a to increase clarity

We have renamed to “Net sediment budget” here and elsewhere.

Line 83: As mentioned above how dependent are the shown ratio for stability on delineation of drainage areas

The error associated with delineation is estimated as 17%, and is now noted (see R1.1 above).

Line 99: Here the calculation for sediment deficit or surplus is discussed, and explained in the methods (line 283). Why is the assumed uniform density chosen that low (150 kg/m³)?

R1.2 The only role of density in this calculation is to couple the local sea-level rise rate (SLR) with the measured sediment flux to yield a “minimum” sediment mass required for survival. The UVVR is independently correlated with sediment flux ($r^2 = 0.77$) and SLR ($r^2 = 0.1$), so we need to assign a density to convert SLR to a sediment mass requirement. We chose a low density for the sediment budget calculation in order to provide a conservative estimate of the sediment deficit or surplus (net sediment budget), and to factor in possible future changes in autochthonous sediment deposition. Given the spatio-temporal uncertainties in density, we decided to choose an arbitrary density for all sites.

We have now used a more robust approach for minimum and mean density. Given the lack of comprehensive data for bulk density at all sites, we have used the comprehensive database from Morris et al. (2016) to provide more defensible values for net sediment budget and lifespan. We use the minimum value (159 kg/m³) in the deficit calculation to provide a conservative estimate on sediment requirements, and the mean value for the lifespan to represent the possible mass of material stored within marsh plains themselves. We also apply the wide variability of +/- 214 kg/m³ to assess the uncertainty associated with the choice of density in the lifespan calculation. This is now detailed on lines 178-185:

“Using a comprehensive compilation of density data⁴⁶, we computed the mean and standard deviation of all brackish and salt marsh sites, which yielded a density of 373 +/- 214 kg/m³ (i.e. average minimum density of 159 kg/m³). For the budget calculation, we use the minimum value across all sites due to high spatial variability in surficial sediment properties even within individual complexes⁴⁷, and to generate a conservative estimate of marsh survival given uncertainties over future autochthonous deposition (which would tend to yield a low density). In this case, the density is required to convert the sea-level rise rate to a representative potential accretion.”

With regards to lifespan (lines 193-196):

“... ρ_{mean} is a representative mean dry bulk density. We use the mean value in this case to represent the density contained within the remaining marsh plain, as opposed to a future potential deposition used in the sediment deficit calculation.”

Line 106: replace tide by tidal

Revised.

Line 115: "flood-ebb SSC a better predictor of sediment flux across all sites", I think the authors miss a chance here to utilize the flood-ebb SSC as a proxy for the morphodynamic equilibrium of the systems, during which following Lanzoni 2012 long-term residual transport short be cancel each other out

We have added this on lines 129-131:

"Prior work also indicates that tidal systems approach morphodynamic equilibrium when flood-ebb SSC differentials and residual sediment approach zero³⁶."

Line 127: needs a reference to be added

We have added a reference to Fagherazzi et al (2013), which states the case that sediment budgets are the most important indicator of marsh trajectory.

Line 134, 269, 297-300: see comment Line 83 → to make this claim delineation methods need to be further explained

The uncertainty analysis surrounding UVVR and delineation is now detailed (see R1.1), and indicates that even with these errors, our claims are substantiated by the analysis. Specifically, all sites still demonstrate a net sediment deficit, and there is still a strong correlation between UVVR and the sediment budget.

Line 275: maybe a short explanation on the measurements would be useful

R1.3 We have expanded the flux methods, with a new supplementary table detailing the environmental conditions (see E1.1 above).

"Briefly, autonomous sensors measuring vertical profiles velocity and near-bottom turbidity were deployed on the channel seabed. Periodic surveys of cross-sectional velocity distribution, channel area, and sediment concentration were used to convert these proxy values to cross-sectionally representative values⁴. Turbidity was converted to suspended-sediment concentration using collected water samples at individual sites, following standard methods."

Line 283: Does local sea level include subsidence?

Yes, as noted from the source. This has been added on lines 176-177:

"... and SLR_{local} is the local rate of sea-level rise (m/y), including vertical land motion."

Reviewer #2 (Remarks to the Author):

The authors then convert the observed net in- or out-flux of sediment to a long-term average annual flux by extrapolation using the average elevation of the marsh surfaces as obtained from USGS national elevation data and local tide-gauge station data (to derive net in- and out-flux for all tides that flood the marsh surface over the period of a year).

To clarify, our sediment flux measurements are not derived from elevation of marsh surfaces and local tide-gauge station data. All but two studies considered here used USGS protocols for water and sediment flux measurement. These involve deployment of autonomous acoustic and optical sensors, vertical and horizontal profiling of velocity, and intensive water sampling. There is little sediment flux literature available that used methods with this level of intensity. Also note that the integrative sediment flux represents sediment imported/exported to-and-from the entire complex, not just the flooded marsh surface. The elevation of the marsh is not utilized in the flux or budget calculations. We have added detail to the methods, and a supplementary table demonstrating the intensive nature of these studies (see response R1.3 above).

Perhaps the biggest single problem with this study is the fact that it uses a static metric (vegetated versus unvegetated marsh area (UVVR)) as a key variable for the prediction of future marsh sedimentary status by

way of time-space substitution (i.e. using the static metric for a range of marsh systems, correlating this against measured net sediment flux, and basing the argument of the future prediction of an individual marsh's life span from UVVR on the outcome of this correlation). Using this method, the authors thus assume that the UVVR represents a direct response to the net sediment flux as measured at the sites during a specific 8-10 month period of a particular year. This seems to me a fundamentally flawed argument, as (a) UVVR is the result of many years of vegetation growth/decay on the marsh surfaces (and there are likely to be significant lags in the system between the processes that drive a change in UVVR and the resulting UVVR itself), and (b) it is known that net sediment fluxes may contain significant inter-annual variability (i.e. an individual 8-10 months of turbidity measurements as used here cannot be used as an integrative measure of sediment budgets over the time scales over which a particular vegetation cover ratio (UVVR) results; this is certainly the case for macrotidal systems (see point 2 below)).

R2.1 Firstly, we appreciate the insight of Reviewer 2 with regards to the time-space substitution, which we did not clearly elucidate in the manuscript. We have made this clear on lines 221-223:

“The range of UVVRs between the 8 sites allows for a time-space substitution, with the implication that sediment export increases as a marsh converts to open water, and UVVR increases.”

Secondly, we agree that the complexity and timescales involved in the creation of open water areas in marshes and the limited duration of sediment flux measurements make the mechanisms behind this correlation seem uncertain. However, recent work (Mariotti, 2016) clearly shows that ponding and open-water expansion can be directly linked to sediment budgets. Furthermore, idealized biogeomorphic modeling (now included in the manuscript as Supplemental Figure 2, and in Methods) shows that the UVVR, sediment supply, and sea-level rise are linked at steady, quasi-equilibrium time-scales. Our work is empirical evidence that these relationships exist despite the limited duration of our observations. On lines 89-93, and in Supplementary Figure 1:

“Idealized numerical modeling of hydrodynamics and vegetated marsh plains³⁰ shows that UVVR increases with decreasing sediment supply and increasing sea-level rise, with stability at UVVR ~ 0.13 (Supplementary Figure 1). Both observations and models indicate the existence of a biogeomorphic tipping point in the capability of a marsh complex to counter changes in external forcing.”

Thirdly, while there can be large interannual variability in fluxes, the datasets used here span a wide spectrum of forcing conditions (as noted on lines 168-173 and in Supplementary Table 1, as well as the original publications).

“Deployments captured over 100 tidal cycles, and several high water events from both extreme spring tides as well as frontal passages (Supplementary Table 1). Additionally, most of the wetland complexes were constrained by topographic controls including uplands and hardened structures. The two sites with the largest tide range were constrained on all edges by roads and uplands (sites SB and OG, respectively). This reduces the uncertainty involved with unchannelized flow at the highest tides.”

The literature upon which these measurements are built have demonstrated the temporal variability in fluxes due to storms and tides (e.g. Rosencranz et al., 2015; Ganju et al., 2013). The key component of these studies is that sediment source is the most important indicator of the sediment flux climate. Once the most frequent forcing conditions have been observed, and the sediment flux mechanisms diagnosed, the source can be inferred. This is central to the flood-ebb SSC metric, which captures the location of the dominant sediment source. Our prior work (Ganju et al., 2013; Ganju et al., 2016) supports these claims.

Lastly, it is important to note that both the UVVR and sediment budgets are intended as “metrics”. As we state in the manuscript on lines 77-79,

“The correlation between UVVR and sediment budget is not necessarily causal; it is a convergence of two integrative indicators that bundle unseen processes across the landscape, which are difficult to measure independently.”

The correlations presented here support the idea that a short-term, sediment-transport based metric is a proxy for longer-term, deteriorative processes. Present-day sediment transport responds to the present-day geomorphic, hydrodynamic, and sedimentary environment of the marsh. The UVVR and the associated implications of open-water conversion are strong controls on those parameters. Recent literature regarding ponding and open-water conversion, cited in the manuscript, strongly supports our claims and our methods are as robust as possible, given the difficulties of continuously monitoring sediment fluxes.

The second most important problem with this study is the assumption that surface elevation change of the marsh complex can be calculated simply through assuming (a) that the bulk density of existing, deposited sediments, is the same as that of the newly deposited sediments measured through the turbidity sensors used in this study to derive sediment fluxes, and (b) uniform bulk density of the sediment that is imported or exported over the annual time-scale.

The manuscript does not calculate surface elevation change. We calculate the mass within the marsh system, and estimate the duration over which that mass would be depleted by the current sediment flux environment. The reviewer is correct that the choice of density controls this timescale, so we have performed a more robust analysis of lifespan, using a range of densities from Morris et al. (2016) (see R1.2 above). We have noted that this lifespan is affected by dynamic changes in sea-level rise and UVVR on lines 103-106:

“Future changes in the rate of open-water conversion, sea-level rise, and climatic factors may modulate sediment export and alter the lifespan prediction. As marshes lower in the tidal frame due to sea-level rise, there may be an increased propensity to trap more sediment and partially offset a sediment deficit.”

D. Appropriate use of statistics and treatment of uncertainties

In addition to the concerns expressed above around the extrapolation of suspended particular matter measurements from 8-10 month to average annual rates, there are other particular instances where further detail and error/uncertainty assessment is required:

Line 297: The vertical accuracy of this data is, at best, 8 cm. This raises concerns around the accuracy of the estimates of elevation relative to mean sea level, as this is used to derive 'life span' based on the computed net sediment flux and marsh area.

We have performed a comprehensive assessment of error associated through use of the NED by recalculating using recently available topographic bare-earth LiDAR at two of our sites (Reedy and Dinner Creeks). The errors in area and UVVR, while large (25% and 17% respectively), do not negate the strong correlation between sediment budget and UVVR (Supplementary Figure 2). We have added error bars to Figure 2a and a detailed description of our uncertainty analysis (see R1.1 above).

Line 301: The authors mention the horizontal resolution but there is no mention of vertical accuracy or resolution, both are critical to the estimation of 'life span' (see above)

The error in elevation arising from use of the NED (assuming the LiDAR dataset is “correct”) was 5.5%. Due to the greater uncertainty in density (+/- 214 kg/m³), we have propagated this error to the calculation of lifespan. Figure 2b now contains error bars and Supplementary Figure 2 now shows the robustness of the correlation (see R1.2 above).

Line 306: It would be useful to know what the potential error is that is introduced by this adjustment.

We believe the error analysis above is a more liberal estimate and will overwhelm any error introduced by this adjustment.

The authors state (line 298-301) that "The boundaries for each study area were identified by the surface slope that drains the marsh water through the sediment flux measurement cross-section. Hydrology tools in ArcMap Spatial Analysis toolbox were used to identify these hydrologic units based on the ~3-m resolution NED." Given the sensitivity of marsh surface flows, in particular macro-tidal systems, to very small (< 10 cm) differences in elevation and to wind conditions during the time of inundation, the methods applied in this paper are, at best, only applicable to very clearly topographically constrained, micro-tidal, marsh systems. It is not clear from the text whether the marsh systems used in this paper are topographically constrained in this way or whether tidal exchange on large spring tides could lead to significant errors in the sediment budgets computed from the single point turbidity sensor data, as has been suggested by thorough studies of sediment budgets in macro-tidal systems nearly 20 years ago (e.g. Murray and Spencer, 1997).

R2.2 Agreed; we have changed the title (and elsewhere in the paper) accordingly to indicate that this paper is focused on microtidal marshes. Nonetheless, microtidal marshes are the most vulnerable to sea-level rise (French, 2006), and represent a sizeable fraction of marshes in the United States and elsewhere (see Kearney and Turner, 2016). We have added these references to the manuscript. On lines 37-39:

"Microtidal marshes are the most vulnerable to future sea-level rise, given their location in the tidal frame and increased dependence on inorganic sediment supply relative to macrotidal systems^{11,12}."

We have added details about topographic constraints the sites in Supplementary Table 1, as well as on lines 169-173:

"Additionally, most of the wetland complexes were constrained by topographic controls including uplands and hardened structures. The two sites with the largest tide range were constrained on all edges by roads and uplands (sites SB and OG, respectively). This reduces the uncertainty involved with unchannelized flow at the highest tides."

E. Conclusions: robustness, validity, reliability

The key issues around the validity of the approach of the study have been outlined under 'D' above. In addition, the following issues somewhat reduce the robustness of the conclusions:

1. The tidal range within the marsh systems used within this analysis is dominated by micro-tidal systems (6 out of the 8 sites used), with the other two being barely meso-tidal systems (maximum tidal range 2.1 to 2.5 m). If all sites were classified based on mean tidal range, all would classify as micro-tidal. At the very least, the conclusions must thus be stated as being limited to micro-tidal marsh systems only. There are in fact many reasons why the results of this analysis can be expected to be vastly different for meso- to macro-tidal systems, e.g.:

We have revised throughout to indicate that this study only applies to microtidal marshes (see R2.2 above).

a. Meteorological conditions play an important role in controlling tidal fluxes of sediment (see e.g. Manning and Bas, 2006; Uncles and Stephens, 2010), and during over-marsh water depths, flow above the vegetation canopy can be significantly correlated with decadal-scale climatic oscillations affecting sea level fluctuations (e.g. Philips and Crisp, 2010; Philips et al, 2013).

While true, the underlying dependency on sediment source is the same. The processes that cause runaway open-water expansion and sediment export (e.g. a sediment deficit) apply equally. Even if flow is dependent on decadal-scale forcing, the source of sediment is the critical variable that determines net sediment flux. We have noted that most of our sites are controlled either topographically or by

hardened structures, including the two with the highest tide ranges (sites OG and SB, see R2.2 above).

b. Tidal flat - salt marsh transitions in meso- to macro-tidal systems, particularly on open coast settings, are often much more dynamic and responsive to variations in meteorological conditions and wave approach angles and energies; including the possibility for marsh edge erosion resulting from the occurrence of individual extreme events. Such events can conceivably cause preferential tidal flat lowering and cliff formation with a runaway effect through reflected wave energy at the cliff causing marsh edge retreat, rather than any alteration in sediment supply or net sediment flux (e.g. Kriby and Kirby's 2008 work on the climatological controls of tidal flat elevations).

R2.3 We agree that an unchannelized, unponded fringing marsh, only experiencing edge erosion might display a low UVVR with a net sediment budget close to zero (or a deficit only due to SLR). But this seems like a relatively rare case, given the ubiquity of channelization and ponds on marsh plains. If there is an external source of sediment, the system can maintain quasi-equilibrium as estuarine/channel bottoms and flats maintain elevation. Redfield (1972) stated this eloquently, indicating that marsh banks can heal themselves if there is a stable supply of sediment to keep the system in equilibrium. More recently, the body of work from Fagherazzi, Mariotti, and others supports the idea that changes in sediment budget are the dominant factor controlling marsh trajectory. The erosion of marsh banks, with no external sediment input, would result in an increasing UVVR and net sediment deficit (assuming some amount of sea-level rise), in agreement with our results.

Additionally, Leonardi et al. (2016) suggest that individual extreme events are not the dominant drivers of marsh erosion, but rather storms with return intervals ~ 2.5 mo.

Leonardi, N., Ganju, N.K. and Fagherazzi, S., 2016. A linear relationship between wave power and erosion determines salt-marsh resilience to violent storms and hurricanes. Proceedings of the National Academy of Sciences, 113(1), pp.64-68.

Given the channelization at all sites, we have reworded to indicate that our relationship holds specifically for channelized marshes, on lines 142-145 (italicized here for emphasis):

“The relationship between sediment budget and UVVR presented here is direct observational evidence of the link between open-water conversion processes and sediment transport, and holds across a geomorphic and climatic spectrum of *channelized*, microtidal marshes.”

2. All marsh systems included in the analysis are North American and found within very limited biogeographical zones (see Figure 1); the response of such marshes to all external drivers mentioned in the paper as well as their internal biological dynamic is fundamentally different to systems elsewhere in the world (see e.g. the diverse and geomorphologically complex and varied systems of the North Atlantic region (see e.g. Allen, 2000), and those in Asia and in the sub-tropical regions of the world), as differences in vegetation composition alter the above- to below-ground biomass ratios, interaction with tidal and wave flows, sediment capture efficiency, not to mention their response to elevated CO₂ and altered atmospheric temperatures as a result of climatic changes that are to be expected over the sea-level rise time-scales considered here.

As noted in the response E1.1, we can only use datasets collected under similar methods for flux estimation. This limited our study to sites where such data have been collected and made available. Nonetheless, these sites are quite dissimilar in terms of climate, surface lithology, and vegetation, now detailed in Table 1. Despite these differences, the relationship between UVVR and sediment budget is robust.

3. The study assumes that sea level rise as derived from past tide gauge data records can simply be extrapolated for the purpose of predicting marsh sedimentary status into the future. It is likely, however, that sea level rise rates are non-linear over time into the future.

R2.4 This is true, and we have revised to indicate that future sea-level rise rates will alter these predictions (lines 103-106).

“Future changes in the rate of open-water conversion, sea-level rise, and climatic factors may modulate sediment export and alter the lifespan prediction. As marshes lower in the tidal frame due to sea-level rise, there may be an increased propensity to trap more sediment and partially offset a sediment deficit⁵.”

F Suggested improvements: experiments, data for possible revision

The idea behind this paper is a very attractive and justifiable one, however, the data base used to approach it and the analysis is not (yet) scientifically robust and rigorous enough. I suggest the authors either:

(a) rewrite the paper as a rather more geographically restricted study, making clear that the conclusions are restricted to a particular type of marsh system that is not representative of salt marshes more generally/globally and addressing the non-linearity that is likely involved in marsh response to drivers such as sea-level rise and accompanying climatic and sediment supply changes over decadal time-scales; or

We agree with the assessment that this study may only be applicable to microtidal salt marshes, and have revised accordingly (see R2.2). However, given our substantial assessment of errors (which do not change the underlying relationship), as well as the newly included idealized modeling that supports our claims, we believe the analysis is robust and rigorous. We have demonstrated a strong correlation between two variables that have independently been postulated as marsh trajectory metrics, and we provide evidence showing that recent idealized modeling work regarding open water conversion and sediment budgets support our empirically based claims. We have acknowledged how future changes in sea-level rise and climate will affect our calculations (see R2.4).

(b) broaden the study to include data from marsh systems with vastly different tidal ranges, sediment and vegetation types, and re-analyse those data with a view to establishing the broad relationships they are searching for.

As indicated above, this is not possible given the paucity of sediment flux data that conform to the standards of the data presented here (see E1.1 above).

However, given the complexity of salt marsh system functioning in a wide variety of climatological and sedimentological settings, the three-dimensional nature with which sediment is introduced and exported from marsh systems, and, furthermore, the expected complexity of marsh system response to non-linear sea level and external tidal sediment loading forcing over decadal time-scales, it would be surprising if the method that the authors are suggesting would yield sensible results, certainly at the larger scale implicit in option (b).

Given the body of modeling work that supports the relationship between sea-level rise, open-water conversion, and sediment budgets, we disagree that our empirical results are surprising. They are expected given the mechanisms responsible for marsh loss and sediment export. The novelty of our findings is the quality of the sediment flux measurements and their relationship to the UVVR. We believe that the additional error analyses and inclusion of idealized numerical modeling strengthen our conclusions.

G References: appropriate credit to previous work?

There seems to be a tendency for over-reliance on the authors' own publications and on US literature. There is a wider body of literature out there that informs on the complexities that are currently not addressed... (see below).

The original submission was limited to 30 references. We have added references as appropriate to this revision, including Murray and Spencer (1997) and French (2006).

LINE BY LINE COMMENTS to support the above statements:

Line 4: vulnerability to what? (unclear sentence)

Revised to “vulnerability to destructive forces”

Line 9: this statement is not really true, as it is extremely critical whether the surplus arises from high vertical accretion and is accompanied by edge erosion...

The case mentioned by the Reviewer represents a recycling of internal material, and would represent a deficit in light of sea-level rise. See R2.3 above with regards to wave-induced edge erosion.

Line 11: No, it can also be addressed through decadal scale annual accurate elevational surveys and tracking of morphological marsh and tidal flat change...

One would also need to accurately survey the channels as well. In any case, that type of measurement gives no indication of the mechanisms responsible for sediment transport; we have revised to include “associated transport mechanisms” on lines 10-11.

Line 12: This is rather geographically restricted and very different vegetation cover to, e.g., NW Europe (reference 5, with which this is contrasted, includes NW European marsh systems, so this comparison is not quite justified...)

Kirwan et al. (2016) makes no distinction in their paper between US and European marshes, so we assume that their results apply to US marshes.

Line 49: What vegetation types were covered? These seem very different to marshes elsewhere in the world...

Vegetation has been added to Table 1; vegetation and climatic type is quite different between these sites. *Spartina alterniflora* is native to the Atlantic but has now invaded the Pacific as well as China and Europe (Li et al., 2009, Vasquez et al., 2006); *Distichlis* is common throughout North America (Hauser, 2006).

Li, B., Liao, C.H., Zhang, X.D., Chen, H.L., Wang, Q., Chen, Z.Y., Gan, X.J., Wu, J.H., Zhao, B., Ma, Z.J. and Cheng, X.L., 2009. *Spartina alterniflora* invasions in the Yangtze River estuary, China: an overview of current status and ecosystem effects. *Ecological Engineering*, 35(4), pp.511-520.

Vasquez, E.A., Glenn, E.P., Guntenspergen, G.R., Brown, J.J. and Nelson, S.G., 2006. Salt tolerance and osmotic adjustment of *Spartina alterniflora* (Poaceae) and the invasive *M haplotype* of *Phragmites australis* (Poaceae) along a salinity gradient. *American Journal of Botany*, 93(12), pp.1784-1790.

Hauser, A.S. 2006. *Distichlis spicata*. In: *Fire Effects Information System*, [Online]. U.S. Department of Agriculture, Forest Service, Rocky Mountain Research Station, Fire Sciences Laboratory (Producer). Available: <http://www.fs.fed.us/database/feis/> [2016, August 25].

Line 51: It is not clear at what spatial resolution the measurements were obtained. There are problems around point-sampling (see also Murray and Spencer (1997)) that are not addressed here.

We have added detail regarding the cross-sectional profiling performed in these studies. These measurements are collected following robust protocols that capture cross-sectional variability in the channel (see E1.1 and R1.3, and Supplementary Table 1).

Line 52: Again, Murray and Spencer (1997) show that this sort of 'scaling' is very likely to lead to significant sediment budget errors...

While the scaling may be prone to error, the idea of the channel-based sediment budget as a metric is nonetheless valid (Ganju et al., 2013). Also note that Murray and Spencer sampled 22 tidal cycles; our shortest dataset covers over 107 tidal cycles and captures multiple storm system passages (Ganju et al., 2013). Perhaps the most important error associated with the scaling is the calculation of lifespan (which is not technically a metric), we have noted this on lines 198-200:

“Scaling errors in the sediment flux calculation (e.g. using several months of data extrapolated to decadal timescales) would cascade to this lifespan calculation.”

Line 55: It seems not accurate to say that the net sediment budget as measured in terms of tidal fluxes is the sediment required to offset sea-level rise; the authors themselves state earlier that organically produced sediment can contribute significantly... such sediment is not captured with this approach here...

This is correct, and therefore the choice of a low density was designed to capture autochthonous deposition. The lower the value, the more we account for locally produced, organic material. We have revised to indicate that the chosen value represents a conservative estimate that accounts for autochthonous production (see R1.2 above).

Line 61: the authors claim that 'loss of vegetated marsh [...] reduces the net trapping potential of a marsh complex by liberating sediment from the substrate'. This is highly unlikely, at least over short time-scales and is highly dependent on the exposure of the unvegetated marsh surface (i.e. where it is located relative to incident wave approach). Spencer et al. (2015) showed that even storm wave conditions failed to liberate sediment from surfaces from which above-ground biomass had been removed, as root and organic matter within the soil substrate prevented re-suspension of sediments. At the very least, this introduces significant inter-annual (and potentially decadal) time-lags before sediment can be mobilised...

We have reworded to indicate “long-term net trapping potential”, and “exposing the substrate” rather than “liberating sediment” on lines 67-70.

“The loss of vegetated marsh, both on the plain and fringes, reduces the long-term net trapping potential of a marsh complex by exposing the substrate and decreasing the momentum extraction and wave attenuation by stems²⁷.”

Line 62: this alone, however, is not enough to drive a net loss in vegetated marsh surface - in some NW European systems, such spatially differentiated erosion leads to accretion elsewhere in the marsh and to the maintenance of the marsh sediment budget in spite of marsh loss at the marsh margins (e.g. discussion in Wolters et al. 2005). Where manmade landward barriers to marsh growth are removed, field evidence also suggests that rapid landward and upward growth of marshes can occur and more than balance seaward fringe erosion (e.g. Spencer et al. 2011):

Wolters, M., Bakker, J. P., Bertness, M. D., Jefferies, R. L., & Möller, I. (2005). Saltmarsh erosion and restoration in south-east England: Squeezing the evidence requires realignment. *Journal of Applied Ecology*, 42(5), 844-851. doi:10.1111/j.1365-2664.2005.01080.x

Spencer, T., Friess, D. A., Möller, I., Brown, S. L., Garbutt, R. A., & French, J. R. (2011). Surface elevation change in natural and re-created intertidal habitats, eastern England, UK, with particular reference to Freiston Shore. *Wetlands Ecology and Management*, 20(1), 9-33. doi:10.1007/s11273-011-9238-y

See R2.3 above. Additionally, the sites mentioned in the references above demonstrate extensive channelization, and our results suggest a sediment deficit would lead to intertidal flat erosion, channel deepening, marsh slumping, and an increasing UVVR within a pre-defined marsh area.

For example, Wolters et al. (2005) states: “Sediment budgets are also important in determining the effects of land claim and dredging on salt marsh erosion. Where sufficient sediment is available, land claim may be followed by increased accretion in front of the new embankment and silting up of the estuarine channels, decreasing the tidal prism and current velocities (Pye 2000). However, when the cross-sectional area of the entrance channel is not affected by land claim, tidal range and current velocities will increase, and the remaining salt marshes are likely to experience lateral erosion of the seaward edge and internal dissection as a result of extension and widening of the creeks.” This supports our analysis of sediment budgets and UVVR.

We now explicitly point out the importance of landward transgression on lines 23-25 and 33-35.

“Salt marshes provide critical ecosystem services but are perceived to be increasingly vulnerable to sea-level rise⁶, sediment starvation⁷, eutrophication⁸, and limits on landward transgression⁵.”

“Physically, whether salt marshes aggrade or retreat is a function of external sediment supply from the watershed or marine end-members² as well as the possibility of landward transgression.”

Our findings are intended to show how open-water conversion of a given, pre-defined marsh complex is related to the sediment budget, not to assess how wetlands can migrate in response to seaward erosion. We have noted the caveat concerning pre-defined areas on lines 153-156:

“Total sediment fluxes (Q_t) from prior studies were scaled to yield yearly fluxes, and normalized by total marsh complex area ($A_t = A_v + A_{uv}$; where A_v and A_{uv} are vegetated and unvegetated area components of a pre-defined complex, respectively) landward of the flux measurement site....”

Line 64: but elsewhere (see previous comment), it is suggested that this process can easily be counter balanced by landward migration of the marsh, space permitting... Also, open water only favors marsh erosion by water currents where those currents are strong enough to erode the sediment - and, as tidal currents are low, that is only the case where waves are of a magnitude high enough to cause erosion - a situation that can only be generated if the individual patches of unvegetated marsh are large enough to allow wind to generate sufficiently high waves over sufficient fetch distances. The study presented here does not take account in any way of the dimension/distribution of unvegetated areas within the marsh complex. It is simply not true that 'open water ultimately favors marsh erosion' (line 63).

Sediment transport in channels, flats, marsh faces, and ponds is always occurring at some level. Ultimately, the sediment supply gradient governs the sediment budget, regardless of the strength of the currents or waves. And, given sea-level rise, a system with no currents or waves will experience a net inorganic sediment deficit if concentrations in the water column are low.

Recent work from Mariotti (2016) shows how small ponds form on the marsh plain and cause permanent marsh loss, depending on external inorganic sediment supply, and in the absence of waves and erosive forces (e.g. mainly slumping or soil creep). We have added text to this effect on lines 70-73.

“In microtidal, sediment-starved settings, increased open water ultimately favors marsh erosion, which in turn increases the unvegetated area, in a positive feedback that deteriorates the entire marsh²⁸. In certain cases, ponding on the marsh plain can result in permanent marsh loss through slumping and soil creep, with runaway erosion preventing recovery²⁸.”

Line 66: yes, but this is all too frequently constrained by man-made structures on the landward margin; where such structures are not present, marsh expansion on the landward margins is made possible and can maintain overall marsh area even where seaward margin erosion is taking place

Not clear which statement is being referred to here. We have indicated in the revision that we are considering pre-defined marsh complexes in this assessment, and the capacity for landward migration is important but not germane to this analysis.

Line 114: ... but only if one assumes that the geometric context which determines the horizontal constraints to tidal flow around the marsh remains fixed over the time-scales over which predictions are made... within the 10-300 year time scale, that is likely to be a flawed assumption...

Again, sediment source is the critical parameter. If the only reliable source of sediment is from marsh edge erosion, this statement is correct over timescales when no other source arises (e.g. riverine input, marine input).

Line 294: This reference is based on micro-tidal marsh systems only.

Revised title to “microtidal”

Line 295: again, this applies to micro-tidal marsh systems only.

Revised title to “microtidal”

Line 297: The vertical accuracy of this data is, at best, 8 cm. This raises concerns around the accuracy of the estimates of elevation relative to mean sea level, as this is used to derive 'life span' based on the computed net sediment flux and marsh area.

As shown in the methods section, use of bare-earth LiDAR topography yielded mean elevation error of 5.5%. Due to the substantially larger error associated with the choice of density, we have only propagated that error to the lifespan calculation and Figure 2b (see R1.1 above).

Line 301: The authors mention the horizontal resolution but there is no mention of vertical accuracy or resolution, both are critical to the estimation of 'life span' (see above)

Following uncertainty analysis above, errors for vertical uncertainty are quantified but not used due to the larger error in choice of density.

Line 306: It would be useful to know what the potential error is that is introduced by this adjustment.

We believe the errors propagated for density, area, and UVVR trump this error (see R1.1 above).

Reviewers' comments:

Reviewer #1 (Remarks to the Author):

I have read the revised version of the manuscript in detail and with great interest. I consider that the Authors have covered all the questions and comments that were posed by myself. I have also read the response to the comments and requests for change of the other reviewer and think the paper greatly benefited from those suggestions especially to proposed shift in focus towards micro-tidal systems.

My original review was already quite positive and the Authors have satisfactorily addressed my comments in the revised manuscript. Below I have list some minor comments regarding the response to questions raised by the other reviewer, which in my opinion still require some attention:

E1.1: Although a broadened study undoubtedly would increase the impact of the manuscript and result in a stronger case, the authors by focusing on microtidal salt marshes increase the accuracy of their arguments, specifically the discussed relevance of the sediment fluxes through channels in combination with UVVR ratios.

R2.1: Although the presented data on sediment measurements in this manuscript are unprecedentedly detailed the criticism of how representative they are on the long-term and therefore their relevance on long-term marsh development has not yet been properly addressed. Especially how do measured sediment fluxes or hydrodynamic forcings compare between years? It is reported that the 8-10 month measurement period included a representative variety of magnitudes of forcing, is this also true on a longer time scale? I am aware that this data might not be available in the same detail as the presented measurement.

I am also not sure how the idealized model can address this shortcoming. In my opinion the idealized modeling (from the Venice lagoon) is described insufficiently, which makes it difficult to determine its value for this manuscript. Therefore I think it should be removed.

R2.2: I appreciate that the authors included a more in depth analysis on their assumptions on bulk density and its caused errors, which were added in Figure.2. However I am wondering whether the uncertainty visible in Figure.2 should not also be included in table.1

R.2.4: I appreciate the author's effort to address the brought up non-linearity in sea level rise and its impact on the marsh's future sedimentary status. I am however wondering whether it should be restricted to certain ranges. Since if due to accelerated sea level rise and the marsh becoming lower in the intertidal frame the system might shift from micro- to mesotidal, sediment fluxes determined in channels potentially become less representative.

Reviewer #2 (Remarks to the Author):

The revisions to this manuscript have taken (almost) all reviewer comments on board and I believe that the manuscript is now much strenghtened and less likely to be mis-interpreted by the reader. The changing of the title to refer to micro-tidal marshes is a much more accurate reflection of the content.

I greatly appreciate the authors making the effort to quantify the potential error terms arising from uncertainties in the way in which the sediment measurements cascade into budget estimates and those then translate into marsh lifespan. I still have some concerns about the use of the ArcGIS watershed tool, but I feel that the authors have added sufficient transparency (and references) to their methods to allow the reader to evaluate these errors further.

The approach to use a conservative estimate of bulk density is also much appreciated and points a clear way forward as to how further studies might deal with uncertainty in light of the need to provide estimates of marsh evolution for coastal management applications.

In their answer to Reviewer 2, R2.1, the authors still seem a little too general and specify too little that their statements are really specific to microtidal settings and highly channelised marsh systems (e.g. on lines 89-93: please change the concluding sentence "Both observations and models indicate... " to "micro-tidal marsh complexes").

I do not follow the authors justification for use of flux studies in long-term sediment budget estimations. They clearly state that other authors (Rosencranz et al., Ganju et al.) have shown the variability due to storms and tides exist, but then they argue that the greatest determinant of sediment flux is sediment source. There are many systems in which storms occur much less frequently than once per year but play a critical role in marsh sediment budgets (see Cahoon 2006). The authors MUST not ignore this possibility when discussing their results!

[1] D.R. Cahoon, A Review of Major Storm Impacts on Coastal Wetland Elevations. *Estuaries and Coasts*. 29 (2006) 889–898.

Although I appreciate the authors revision of the text on lines 103-106 re sea level rise uncertainty, the potential role of extreme events needs to be stated and it needs to be made clear that this may mean that the short-term sediment-transport based metric used here is not NECESSARILY applicable or at least should be treated with caution - particularly when it is applied over these very long time-scales over which storm magnitude and frequency is likely to drastically change due to climatic changes....The authors refer to Leonardi et al. (2016), but this again is a much more geographically limited study than that of Cahoon (2006) above....

I appreciate the references to the possibility of landward transgression now included.

In addition to my points above, the authors should change the wording in the abstract as follows to just clarify these points:

Line 9: 'sediment surplus causes expansion, while a sediment deficit causes contraction' – this statement is simply too vague and, given the physical disconnect between vertical accretion and lateral dynamics (as also pointed out in reference (3) used by the authors here), is not helpful: expansion and contraction of what? Vegetated surfaces? If so, then I do not think the statement is true for many of the little fragmented but laterally constrained marshes of NW Europe, for example, where sediment budgets may be positive (much sediment is being captured by the marsh surfaces and creeks are not eroding), but the marsh is still retreating laterally due to wave attack, not necessarily with a release of sediment that matches the quantity of sediment deposited on the marsh surfaces. A sediment surplus (if measured with respect to the area defined by the vegetated marsh surfaces) can thus exist even when the marsh extent is slowly retreating (this would be the case for marshes classed in reference (3) in the manuscript as having a very long 'Geological life cycle').

I suggest to replace the sentence on Line 9 with something like this:

"A sediment surplus may indicate vertical growth and/or lateral expansion, a sediment deficit may indicate contraction and/or degradation and 'ecological drowning'".

Line 12: This is not correct: extended tidal-timescale observations of sediment transport are not the only way of quantifying integrative sediment budgets... Rapid advances in LiDAR and other surveying methods are now offering opportunities for the mm accurate mapping of three-dimensional marsh morphology and its change over time. This may ultimately be a method with less error than that involved in the extrapolation of sediment budgets from point measurements within tidal waters....

I would urge the authors to review this sentence to something like:

"...that requires extended tidal timescale observations of sediment transport or the accurate mapping of three-dimensional marsh morphology change over time".

Line 18: This sentence HAS to include reference to the rather gross assumptions made in this extrapolation exercise. I suggest:
"... at current rates of sea-level rise sediment availability."

Reviewer #1 comments:

I have read the revised version of the manuscript in detail and with great interest. I consider that the Authors have covered all the questions and comments that were posed by myself. I have also read the response to the comments and requests for change of the other reviewer and think the paper greatly benefited from those suggestions especially to proposed shift in focus towards micro-tidal systems.

My original review was already quite positive and the Authors have satisfactorily addressed my comments in the revised manuscript.

Thank you for your comments and support.

E1.1: Although a broadened study undoubtedly would increase the impact of the manuscript and result in a stronger case, the authors by focusing on microtidal salt marshes increase the accuracy of their arguments, specifically the discussed relevance of the sediment fluxes through channels in combination with UVVR ratios.

We agree, thank you.

R2.1: Although the presented data on sediment measurements in this manuscript are unprecedentedly detailed the criticism of how representative they are on the long-term and therefore their relevance on long-term marsh development has not yet been properly addressed. Especially how do measured sediment fluxes or hydrodynamic forcings compare between years? It is reported that the 8-10 month measurement period included a representative variety of magnitudes of forcing, is this also true on a longer time scale? I am aware that this data might not be available in the same detail as the presented measurement.

Analyzing the inter-annual variability in hydrodynamic forcings and fluxes at all sites is beyond the scope of this paper, but we acknowledge that this introduces some uncertainty. Following the similar comment of Reviewer 2, we have added text to this effect on lines 180-183:

“There may still be significant uncertainty in the extrapolation of sediment flux estimates given annual and episodic variability. Individual storms with >1 y recurrence intervals can deposit substantial quantities of sediment on the marsh plain and alter long-term trajectory⁴⁰.”

I am also not sure how the idealized model can address this shortcoming. In my opinion the idealized modeling (from the Venice lagoon) is described insufficiently, which makes it difficult to determine its value for this manuscript. Therefore I think it should be removed.

The idealized modeling supports the relationship between sediment supply and UVVR; it is not intended to validate the representativeness of the sediment flux data. We have added more description (italicized) for the idealized model on lines 91-97, and 271-285:

“Idealized numerical modeling of *the biogeomorphic patterns* of vegetated marsh plains *in the Venice Lagoon*³⁰ shows that UVVR increases with decreasing sediment supply and increasing rates of sea-level rise, with stability at UVVR ~ 0.13 (Supplementary Figure 1). *This value is consistent with the UVVR value determined for the actual tidal watershed within the Venice Lagoon.* Both observations and models indicate the existence of a biogeomorphic tipping point in the capability of a microtidal marsh complex to counter changes in external forcing.”

“We used a biogeomorphic model of salt-marsh evolution³⁰ to address the effects of sediment supply and rates of sea level rise on the UVVR. The model uses a simplified treatment of the two-dimensional shallow water equations⁵⁰ to describe marsh hydrodynamics and determines platform evolution *by coupling the Exner sediment balance equation with the solution of an advection-dispersion equation over the marsh platform. Changes in marsh surface elevation are dictated by the balance between erosion, inorganic deposition through settling and particle capture by vegetation, and organic soil production deriving from the competition among different vegetation species.* Numerical experiments were carried out for the tidal watershed of an actual channel network within the San Felice salt marsh, in the Venice Lagoon, Italy. Constant suspended-sediment concentrations of 10, 20, and 30 mg/L (representing external sediment supply) were specified within the channel network, with imposed sea-level rise rates of 3, 5, and 7 mm/y. Net sediment deficit was calculate following Equations 1-2 above. *Different sediment supplies and rates of sea-level rise promoted the formation of different marsh topographic configurations, characterized by the transition of vegetated marsh portions to unvegetated areas, thus influencing the UVVR ratio.*”

R2.2: I appreciate that the authors included a more in depth analysis on their assumptions on bulk density and its caused errors, which were added in Figure.2. However I am wondering whether the uncertainty visible in Figure.2 should not also be included in table.1

We have added uncertainties for the lifespan computation to Table 1. We considered adding uncertainties to the table for the net sediment budget, but the values cannot be expressed as a simple bounds, as the uncertainty is related to error in the area and density, which contribute errors to the SLR-related term and sediment flux term unequally (e.g. the values for site BW are -1.06 + 0.09 or -1.06 - 0.15 kg/m²/y).

R.2.4: I appreciate the author’s effort to address the brought up non-linearity in sea level rise and its impact on the marsh’s future sedimentary status. I am however wondering whether it should be restricted to certain ranges. Since if due to accelerated sea level rise and the marsh becoming lower in the intertidal frame the system might shift from micro- to mesotidal, sediment fluxes determined in channels potentially become less representative.

For such a situation to occur, there would need to be a mesotidal oceanic end-member. Then, for a landward marsh system to shift from microtidal to mesotidal would require a drastic modification in the hydrodynamics of the seaward end of the estuary. It would require a system with excessive tidal attenuation to deepen (via SLR) by enough to eliminate frictional effects. None of these systems is in such a situation where the oceanic end-member is mesotidal and the tide is attenuated to microtidal at the marsh.

Reviewer #2 comments:

The revisions to this manuscript have taken (almost) all reviewer comments on board and I believe that the manuscript is now much strengthened and less likely to be misinterpreted by the reader. The changing of the title to refer to micro-tidal marshes is a much more accurate reflection of the content. I greatly appreciate the authors making the effort to quantify the potential error terms arising from uncertainties in the way in which the sediment measurements cascade into budget estimates and those then translate into marsh lifespan. I still have some concerns about the use of the ArcGIS watershed tool, but I feel that the authors have added sufficient transparency (and references) to their methods to allow the reader to evaluate these errors further.

The approach to use a conservative estimate of bulk density is also much appreciated and points a clear way forward as to how further studies might deal with uncertainty in light of the need to provide estimates of marsh evolution for coastal management applications.

Thank you, we appreciate the contributions you have made to the improvement of this manuscript.

In their answer to Reviewer 2, R2.1, the authors still seem a little too general and specify too little that their statements are really specific to microtidal settings and highly channelised marsh systems (e.g. on lines 89-93: please change the concluding sentence "Both observations and models indicate... " to "micro-tidal marsh complexes").

Revised to add "microtidal". With the title changed in the last revision, it should be clear that these relationships are intended to apply to only microtidal systems.

I do not follow the authors' justification for use of flux studies in long-term sediment budget estimations. They clearly state that other authors (Rosencranz et al., Ganju et al.) have shown the variability due to storms and tides exist, but then they argue that the greatest determinant of sediment flux is sediment source. There are many systems in which storms occur much less frequently than once per year but play a critical role in marsh sediment budgets (see Cahoon 2006). The authors MUST not ignore this possibility when discussing their results!

Although I appreciate the authors revision of the text on lines 103-106 re sea level rise uncertainty, the potential role of extreme events needs to be stated and it needs to be made clear that this may mean that the short-term sediment-transport based metric used here is not NECESSARILY applicable or at least should be treated with caution - particularly when it is applied over these very long time-scales over which storm magnitude and frequency is likely to drastically change due to climatic changes....The authors refer to Leonardi et al. (2016), but this again is a much more geographically limited study than that of Cahoon (2006) above....

While we still maintain that identifying sediment source is possible through short-term measurements, we recognize that infrequent storms can greatly modify the sediment budget. We have added a reference to Cahoon (2006). On lines 180-183:

"There may still be significant uncertainty in the extrapolation of sediment flux estimates given annual and episodic variability. Individual storms with >1 y recurrence intervals can deposit substantial quantities of sediment on the marsh plain and alter long-term trajectory⁴⁰."

In addition to my points above, the authors should change the wording of the abstract as follows to just clarify these points:

Line 9: 'sediment surplus causes expansion, while a sediment deficit causes contraction' – this statement is simply too vague and, given the physical disconnect between vertical accretion and lateral dynamics (as also pointed out in reference (3) used by the authors here), is not helpful: expansion and contraction of what? Vegetated surfaces? If so, then I do not think the statement is true for many of the little fragmented but laterally constrained marshes of NW Europe, for example, where sediment budgets may be positive (much sediment is being captured by the marsh surfaces and creeks are not eroding), but the marsh is still retreating laterally due to wave attack, not necessarily with a release of sediment that matches the quantity of sediment deposited on the marsh surfaces. A sediment surplus (if measured with respect to the area defined by the vegetated marsh surfaces) can thus exist even when the marsh extent is slowly retreating (this would be the case for marshes classed in reference (3) in the manuscript as having a very long 'Geological life cycle'). I suggest to replace the sentence on Line 9 with something like this: "A sediment surplus may indicate vertical growth and/or lateral expansion, a sediment deficit may indicate contraction and/or degradation and 'ecological drowning'".

We agree, revised on lines 6-9 to:

“Marsh sediment budgets represent a spatially integrated measure of competing constructive and destructive forces: a sediment surplus may result in vertical growth and/or lateral expansion, while a sediment deficit may result in drowning and/or lateral contraction.”

Line 12: This is not correct: extended tidal-timescale observations of sediment transport are not the only way of quantifying integrative sediment budgets... Rapid advances in LiDAR and other surveying methods are now offering opportunities for the mm accurate mapping of three-dimensional marsh morphology and its change over time. This may ultimately be a method with less error than that involved in the extrapolation of sediment budgets from point measurements within tidal waters....

I would urge the authors to review this sentence to something like:

“...that requires extended tidal timescale observations of sediment transport or the accurate mapping of three-dimensional marsh morphology change over time”.

This sentence has been removed to comply with the 150-word limit of the abstract.

Line 18: This sentence HAS to include reference to the rather gross assumptions made in this extrapolation exercise. I suggest:

“... at current rates of sea-level rise sediment availability.”

We agree, revised on lines 12-14 to:

“All sites are exhibiting a sediment deficit, with half the sites having projected lifespans of less than 350 years at current rates of sea-level rise and sediment availability.”

REVIEWERS' COMMENTS:

Reviewer #1 (Remarks to the Author):

I have read the revised version of the manuscript with great interest. All reviewer comments have been addressed sufficiently, both strengthening the manuscript and clarifying sections which could have be misinterpreted.

I have no further comments.

Reviewer #2 (Remarks to the Author):

Thank you for addressing all my remaining comments. The material reported in this paper has now been placed into a much more meaningful context and realistic assessment.